

# Modeling of PAHs From Global to Regional Scales: Model Development and Investigation of Health Risks from 2013 to 2018 in China

Zichen Wu[1,2,3], Xueshun Chen[1,2,3]*, Zifa Wang[1,2,3]*, Huansheng Chen[1,2,3], Zhe Wang[1,2,3], Qing Mu[4], Lin Wu[1,2,3], Wending Wang[1,2,3], Xiao Tang[1,2,3], Jie Li[1,2,3], Ying Li[1,2,3], Qizhong Wu[5], Yang Wang[3,6], Zhiyin Zou[1,2,3], Zijian Jiang[1,2,3]

[1] State Key Laboratory of Atmospheric Boundary Layer Physics and Atmospheric Chemistry, Institute of Atmospheric Physics, Chinese Academy of Sciences, Beijing 100029, China

[2] Key Laboratory of Atmospheric Environment and Extreme Meteorology, Institute of Atmospheric Physics, Chinese Academy of Sciences, Beijing 100029, China

[3] University of Chinese Academy of Sciences, Beijing 100049, China

[4] Department of Health and Environmental Sciences, School of Science, Xi'an Jiaotong-Liverpool University, Suzhou 215123, China

[5] Beijing Normal University, Beijing 100875, China

[6] Research Center for Eco-Environmental Sciences, Chinese Academy of Sciences, Beijing 100085, China

*Correspondence to:* Xueshun Chen (chenxsh@mail.iap.ac.cn) and Zifa Wang (zifawang@mail.iap.ac.cn)

**Abstract.** Polycyclic aromatic hydrocarbons (PAHs) significantly impact human health due to their persistence, toxicity, and potential carcinogenicity. Their global distribution and regional changes caused by emission changes, especially over areas in developing countries, remain to be understood along with their health impacts. This study implemented a PAH module in the global-regional nested Atmospheric Aerosol and Chemistry Model (IAP-AACM) to investigate the global distribution of PAHs and the change in their health risks from 2013 to 2018 in China. An evaluation against observations showed that the model could capture well the spatial distribution and seasonal variation of Benzo[a]pyrene (BaP), the typical indicator species of PAHs. At a global scale, the annual mean concentrations are highest in China, followed by Europe and India, with high values exceeding the target values of 1 ng m$^{-3}$ over some areas. Compared with 2013, the concentration of BaP in China decreased in 2018 due to emission reductions, whereas it increased in India and Southern Africa. However, the decline is much smaller than for PM$_{2.5}$ during the same period. The concentration of BaP decreased by 8.5% in Beijing-Tianjin-Hebei (BTH) and 9.4% in the Yangtze River Delta (YRD). It even increased over areas in the Sichuan Basin due to changes in meteorological conditions. The total incremental lifetime cancer risk (ILCR) posed by BaP





only showed a slight decrease in 2018 and the population in East China still faced significant potential
health risks. The results indicate that strict additional control measures should be taken to reduce the
pollution and health risks of PAHs effectively. The study also highlights the importance of considering
changes in meteorological conditions when evaluating emission changes from concentration monitoring.
**1 Introduction**

Polycyclic aromatic hydrocarbons (PAHs) are aromatic compounds with two or more aromatic rings.

PAHs have been categorized as persistent organic pollutants (POPs) by the United Nations Economic
Commission for Europe's (UNECE's) Convention on Long-Range Transboundary Air Pollution
(CLRTAP) (Friedman and Selin, 2012), and they are widely distributed in the environment through
atmospheric transport. PAHs have attracted significant attention in environmental research and risk
assessment due to their persistence, toxicity, and potential carcinogenicity (Chen and Liao, 2006; Shen
et al., 2014). These compounds are generated from both natural and anthropogenic sources (Haritash and
Kaushik, 2009). Volcanic eruption, forest, and prairie fire are the major natural sources of atmospheric
PAHs (Baek et al., 1991). Anthropogenic sources are the most important source of PAHs, including
incomplete combustion of fossil fuels and biomass (Li et al., 2022; Ravindra et al., 2008).

Understanding the sources, distribution, and fate of PAHs is crucial for assessing their impacts on

human health and the environment. Upon emission into the atmosphere, PAHs are redistributed by gas-
particulate partitioning, gaseous-phase reactions, heterogeneous reactions, air-soil exchange, and wet/dry
deposition during long-range transport (LRT, Inomata et al., 2013). Monitoring is the most commonly
used way to investigate the concentration of PAHs in the atmosphere. Due to the high costs of observation
and technical limitations, it is difficult to conduct a long-term and broad regional analysis through
monitoring (Zhen, 2023). Up to now, there are few continuous observations over the major continents at
the same time (Dong et al., 2023). A transport model is an effective tool to simulate the distribution of
PAHs and their LRT, which can greatly enhance our understanding of the distribution of PAHs on a
regional and global scale (Byun and Schere, 2006; Wang et al., 2021).

As recently outlined by Galarneau et al. (2014), several numerical modeling studies have been

reported in the literature. The models that can simulate PAHs include but are not limited to the following
examples, GEOS-Chem (Friedman et al., 2014; Friedman and Selin, 2012), ECHAM5 (Lammel and



Sehili, 2007; Lammel et al., 2009; Lammel et al., 2015; Octaviani et al., 2019), CAM5 (Lou et al., 2023;
Shrivastava et al., 2017), and MOZART-4 (Shen et al., 2014). The horizontal resolutions of these reported
models are primarily at $4°\times5°$ and $2.8°\times2.8°$. Shen et al. (2014) simulated the transport of
Benzo[a]pyrene (BaP), one of the most toxic and highly carcinogenic PAHs, in the global troposphere
based on MOZART-4, and they showed that the model resolution was crucial for the health risks
assessment. Lammel et al. (2015) demonstrated the significant impact of gas-particle partitioning
mechanisms on the atmospheric lifetime, compartment distributions, and LRT of PAHs. The regional
modeling studies focusing on Europe, East Asia, and North America have also been reported, with
horizontal resolutions ranging mainly from 54 km×54 km to 24 km×24 km (CMAQ (Aulinger et al.,
2009; Aulinger et al., 2007; Bieser et al., 2012; San José et al., 2013; Efstathiou et al., 2016), WRF-Chem
(Mu et al., 2018), AURAMS (Galarneau et al., 2014), and CanMETO (Zhang et al., 2011a; Zhang et al.,
2011b; Zhang et al., 2009)). Efstathiou et al. (2016) showed that considering absorption and adsorption
processes can better capture the concentration levels and seasonal variations of BaP. In recent years, the
effect of the heterogeneous reaction process of PAHs on transportation has also been studied. Mu et al.
(2018) developed a new kinetic scheme describing the effects of temperature and humidity on the organic
aerosol coating of BaP and BaP reaction rate. They found that low temperature and low humidity can
significantly increase the lifetime of BaP and enhance its LRT capacity.

However, the resolutions and spatial range differed greatly between these models. Most of the

models are either global or regional. There is a lack of simulation studies focusing on both global and
key regions, making it difficult to investigate a specific focus in a global background in a consistent
manner. Additionally, the resolution of most global models is low, which will further affect the health
risk assessment of PAHs. Furthermore, the up-to-date mechanisms (gas-particle partitioning,
heterogeneous reaction, and air-soil exchange) established for PAHs simulations are not considered in
earlier modeling studies.

China is one of the largest PAH-emitting countries in the world (Inomata et al., 2012; Zhang and

Tao, 2009). High concentrations of BaP have been reported (Bieser et al., 2012; Liu et al., 2014;
Shrivastava et al., 2017; Su et al., 2023). Over the polluted regions in eastern China, annual
concentrations of BaP exceeded 1 ng m$^{-3}$, the target values proposed in the European Union (EU) and
China. To improve air quality and protect public health, the State Council of China promulgated "the



Action Plan on Air Pollution Prevention and Control" (the Action Plan) in 2013. Since then, many studies
have investigated the changes in concentration levels and health risks of conventional pollutants, such as
$PM_{2.5}$ (Feng et al., 2019; Wang et al., 2018; Zhang et al., 2019; Zhu et al., 2021; Wang et al., 2019). Wang
et al. (2019) pointed out that the annual average concentrations of $PM_{2.5}$ in the Beijing-Tianjin-Hebei
(BTH), the Yangtze River Delta (YRD), and the Pearl River Delta (PRD) all decreased by more than 27%
in 2017, indicating that the control measures have achieved remarkable effects and the air quality has
been significantly improved. However, for non-conventional pollutants, such as BaP and other PAHs,
their concentration changes due to emission reduction in China after implementing of policies have not
been quantified. The changes in health risks and the benefits from control measures were not yet assessed.
Considering the aforementioned, we simulated PAHs from global to regional scales by coupling the
key physical and chemical modules associated with PAHs in a global-regional nested atmospheric
transport model. In particular, newly established parametrizations of gas-particle partitioning and
heterogeneous reaction were incorporated into the model. Then the changes in global concentration and
health risks of BaP over China were quantified based on model evaluation against a collected observation
dataset. The study can advance our understanding of global PAHs distribution and regional health risks
and their responses to emission change. The paper is arranged as follows: Section 2 briefly describes the
host model (IAP-AACM), the physical and chemical modules related to PAHs, and the method of
assessing health risks. Section 3 presents the configuration of the model and the observations used in the
evaluation. Section 4 shows the global and regional distributions of BaP concentrations and analyzes the
health risks associated with BaP in China. Section 5 discusses the uncertainty of the model. In Sect. 6,
the main conclusions are summarized.
**2 Model description and development**
**2.1 Description of host model**
The model used in this study is the Atmospheric Aerosol and Chemistry Model developed by the
Institute of Atmospheric Physics, Chinese Academy of Sciences (IAP-AACM) (Wei et al., 2019), which
was developed based on the Global Nested Air Quality Prediction Modeling System (GNAQPMS, Chen
et al., 2015; Wang et al., 2001). IAP-AACM is a 3-D Eulerian transport model that uses a multi-scale
domain-nesting approach to simulate atmospheric chemistry and aerosol processes from global to





regional scales. As recently described by Chen et al. (2015), compared with the traditional multi-scale
modeling methods (Seigneur et al., 2001), the online nesting method uses the same parameters in the
global and regional domains, which avoids uncertainties caused by different boundary conditions, and it
also provides boundary conditions at higher time resolution (Zhang et al., 2012b; Chen et al., 2015), thus
improving the performance of the model at the regional scale.

This model includes emission, horizontal and vertical advection (Walcek and Aleksic, 1998),

diffusion (Byun and Dennis, 1995), dry deposition (Zhang et al., 2003), gaseous chemistry (CBM-Z,
Carbon Bond Mechanism version Z, Zaveri and Peters, 1999), heterogeneous chemistry (Li et al., 2012),
aqueous reactions in clouds, and wet scavenging (Stockwell et al., 1990). It has been successfully and
widely applied to simulate the spatial-temporal distribution characteristics of gaseous pollutants, aerosol
components, and the long-distance transportation of mercury (Chen et al., 2015; Chen et al., 2014; Wei
et al., 2019; Ye et al., 2021; Du et al., 2019). In addition, advanced particle microphysics (APM) has been
incorporated to simulate new particle formation processes and predict the particle number concentrations
at global and regional scales (Chen et al., 2021).
**2.2 Development of the PAH module**

The PAH processes in the IAP-AACM model include gaseous-phase reaction, heterogeneous

reaction, gas-particle partitioning, air-soil exchange, dry deposition, and wet scavenging. The simulated
species include BaP, Benzo[b]fluorathene (BbF), Benzo[k]fluorathene (BkF), and Indeno[1,2,3-
cd]pyrene (IcdP) in the gas and particulate phases (Wu et al., 2024). In this study, we mainly focus on
BaP due to its highly carcinogenic nature and the relatively rich observations.
**2.2.1 Gaseous-phase reactions**

PAHs are degraded through reactions with various atmospheric oxidants such as hydroxyl radical

(OH), nitrate radical (NO$_3$), and ozone (O$_3$) in the troposphere (Lammel and Sehili, 2007). Among these
oxidants, the reactions with OH are considered to be the most important pathway for the removal of
PAHs. The nighttime reaction of PAHs with NO$_3$ is also important in the atmosphere (Keyte et al., 2013).
Therefore, reactions of gaseous-phase BaP with OH, NO$_3$, and O$_3$ are all considered in the model. The
second-order rate coefficients are $5.0\times10^{-11}$, $5.4\times10^{-11}$, and $2.6\times10^{-17}$ cm$^3$ molecules$^{-1}$ s$^{-1}$, respectively
(Inomata et al., 2013; Finlayson-Pitts and Pitts, 2000; Klöpffer et al., 2007).



### 2.2.2 Heterogeneous reaction


In the case of BaP, the heterogeneous reaction with $O_3$ is considered to be the dominant loss
(Finlayson‑Pitts and Pitts, 2000; Efstathiou et al., 2016). Studies have shown that the process of
heterogeneous reaction can be well described by the Langmuir-Hinshelwood mechanism (Kahan et al.,
2006; Kwamena et al., 2007), in which BaP is adsorbed to the surface while the $O_3$ is in phase equilibrium.
The first-order reaction rate coefficient k ($s^{-1}$) of the Langmuir-Hinshelwood mechanism is as follows:

$$k = \frac{k_{max}K_{O_3}[O_3]}{1 + K_{O_3}[O_3]} \tag{1}$$

$$\frac{\partial C}{\partial t} = -K_{O_3}[O_3] \tag{2}$$

Where $k_{max}$ is the maximum rate coefficient, and the value is $0.060 \pm 0.018$ $s^{-1}$. $[O_3]$ is the
concentration of $O_3$ (mol $cm^{-3}$). $K_{O_3}$ is the $O_3$ to surface equilibrium constant ($0.028 \pm 0.014 \times 10^{-13}$ $cm^3$).
In addition, we incorporated a more detailed parameterization (ROI-T) developed by Mu et al. (2018)
based on the Langmuir-Hinshelwood mechanism. The scheme emphasizes the importance of
representing the dependence of degradation on temperature and humidity, when coated by organic
aerosols The first-order reaction rate coefficient $k$ ($s^{-1}$) is given by Eq. (3).

$$k = base + \frac{max - base}{1 + (\frac{xhalf}{[O_3]})^{rate}} \tag{3}$$

Where $base$, $max$, $rate$, and $xhalf$ are all the parameterizations of the heterogeneous reaction,
with specific values shown in Mu et al. (2018). In our study, we coupled these two parameterizations as
two options for $O_3$ degradation by heterogeneous reaction in IAP-AACM. The model results using these
two schemes were compared to analyze the influence of heterogeneous reaction schemes on BaP
concentration. The ROI-T scheme was used as the default in this study.

### 2.2.3 Gas-particle partitioning


The partition of compounds between the gas and particulate phases is parameterized with the gas-
particle partitioning coefficient ($K_P$, $m^3$ $\mu g^{-1}$) (Harner and Bidleman, 1998):

$$K_P = (\frac{[PAH]\_p}{[TSP]})/[PAH]\_g \tag{4}$$

Where $[PAH]\_g$ and $[PAH]\_p$ are the concentrations of PAHs in the gas and particulate phase
($\mu g$ $m^{-3}$), and $[TSP]$ is the concentration of total suspended particles (TSP, $\mu g$ $m^{-3}$) in the atmosphere



(µg m$^{-3}$).
Adsorption onto black carbon (BC) and absorption into aerosol organic matter (OM) are two
important mechanisms of gas-particle partitioning (Odabasi et al., 2006). Therefore, we use the gas-
particle partition coefficient equation to represent these two mechanisms, which was derived by Dachs
and Eisenreich, 2000:

$$K_P = \left[\frac{(f_{OM}MW_{OCT}\delta_{OCT})K_{OA}}{(\rho_{OCT}MW_{OM}\delta_{OM}10^{12})}\right] + [\left(\frac{f_{BC}a_{BC}K_{SA}}{a_{AC}10^{12}}\right)] \tag{5}$$

Where $MW_{OCT}$ and $MW_{OM}$ are the mean molecular weights of octanol and OM phase (g mol$^{-1}$),
$\delta_{OCT}$ and $\delta_{OM}$ are the activity coefficient of the absorbing compound in octanol and OM phase,
respectively. $f_{OM}$ and $f_{BC}$ are the mass fractions of OM phase on TSP and the BC in the aerosol. $\rho_{OCT}$
is the density of octanol (0.820 kg L$^{-1}$). $a_{BC}$ and $a_{AC}$ are the specific surface areas of BC (62.7 m$^2$ g$^{-1}$,
Jonker and Koelmans, 2002) and activated carbon (AC), respectively. In this study, we use the same
assumption as Odabasi et al. (2006) ($MW_{OCT}/MW_{OM} = 1$ , $\delta_{OCT}/\delta_{OM} = 1$, and $a_{BC}/a_{AC} = 1$).

$$logK_{OA} = A + B/(T) \tag{6}$$

Where $K_{OA}$ is the octanol–air partitioning coefficient (temperature dependent). T is the
temperature (K). The values of $A$ and $B$ are 5382 and -6.5, respectively (Odabasi et al., 2006).

$$logP_L = m_L(T)^{-1} + b_L \tag{7}$$

$$logK_{SA} = -0.85logP_L + 8.94 - \log\left(\frac{998}{a_{BC}}\right) \tag{8}$$

Where $P_L$ is the supercooled liquid vapor pressure (Pa). The values of $b_L$ and $m_L$ are 12.59 and
-5252, respectively (Dachs and Eisenreich, 2000). $K_{SA}$ is the soot–air partitioning coefficient (L kg$^{-1}$),
which is a function of $P_L$ and $a_{BC}$ (van Noort, 2003).

**2.2.4 Air-soil exchange**

The semi-volatility and persistence of PAHs allow them to dynamically exchange between the
atmosphere and soil by deposition and re-volatilization from ground surfaces (Semeena and Lammel,
2005). These processes can affect the distribution and long-distance transport of PAHs in the environment.
As described by Hansen et al. (2004), air–soil exchange is parameterized following Strand and Hov
(1996), which is based on Jury et al. (1983). Here, soil (standard soil) is considered to be a homogeneous
layer of thickness $z_s$= 0.15 m, and standard values and chemical properties are provided by Jury et al.
(1983) (Table S1). The differential equation for the change of concentrations in soil and air can be





expressed by Eq. (9) and Eq. (10):

$$\frac{\partial c_s}{\partial t} = \frac{1}{z_s}\left(F_{exc,soil} + F_{wet}\right) - k_{soil}c_s \tag{9}$$

$$\frac{\partial c_a}{\partial t} = -\frac{1}{z_a}F_{exc,soil} \tag{10}$$

Where $C_a$ and $C_s$ are the concentrations of PAHs in the atmosphere and soil, respectively. The $z_a$
is the lowest atmospheric layer depths (m), $F_{wet}$ is the wet deposition flux (mol s$^{-1}$ m$^{-2}$). $k_{soil}$ is the
degradation rate in soil, which is estimated to be $2.2 \times 10^{-8}$ s$^{-1}$ (Finlayson‐Pitts and Pitts, 2000; Klöpffer
et al., 2007; Lammel et al., 2009). The air–soil exchange flux ($F_{exc,soil}$) is given by Eq. (11):

$$F_{exc,soil} = K_{a/s}\left(c_a - \frac{c_s}{K_{soil-air}}\right) \tag{11}$$

$K_{soil-air}$ is the partitioning coefficient between soil and air, which is given by Karickhoff (1981):

$$K_{soil-air} = 4.11 \times 10^{-4} \times \rho_s f_{oc} K_{OA} \tag{12}$$

Where $f_{OC}$ is the fraction of OC in soil and $4.11 \times 10^{-4}$ is a constant with units of m$^3$ kg$^{-1}$. $\rho_s$ is the
density of soil. $K_{a/s}$ is the overall exchange velocity (m s$^{-1}$), which can be estimated by Eq. (13) (Strand
and Hov, 1996):

$$K_{a/s} = \frac{D_G^{air}a^{10/3}(1-l-a)^{-2} + D_L^{water}l^{10/3}K_{WA}(1-l-a)^{-2}}{z_s/2} \tag{13}$$

Where $D_G^{air}$ and $D_L^{water}$ are the air and liquid diffusion coefficient (m$^2$ s$^{-1}$), respectively. $K_{WA}$ is
the water-air partitioning coefficient. The differential equation is solved the ODEPACK
(https://github.com/jacobwilliams/odepack).

**2.2.5 Dry and wet deposition**

PAHs can be removed from the atmosphere and enter terrestrial ecosystems through dry and wet
deposition (Cao et al., 2021). Dry deposition and wet scavenging have been included in IAP-AACM. For
the gaseous species of PAHs, their wet scavenging is assumed to be the same as xylene in the CBMZ
mechanism, which is also an aromatic hydrocarbon like BaP; for the PAHs in the particle phase, these
two processes are treated similarly to that of organic aerosol.

**2.3 Risk assessment**

The incremental lifetime cancer risk (ILCR) is widely used to calculate the risk of human exposure
to PAHs (Nam et al., 2021). The carcinogenic risk of PAHs to humans through different exposure routes



was calculated based on the health risk evaluation model proposed by the U.S. Environmental Protection
Agency (EPA) (Smith et al., 1999).
The national population data in 2013 and 2018 were obtained from the LandScan (Oak Ridge
National Laboratory; database can be accessed via: https://landscan.ornl.gov, last access: 20 January
2024) and re-gridded to $1° \times 1°$ and $0.33° \times 0.33°$ to match the model resolution.

**2.3.1 Daily exposure dose**

Dermal contact and inhalation are regarded as the major routes of human exposure to BaP (Li et al.,
2010; Ma et al., 2020; Zhang et al., 2016). In this study, the health risk for the entire population and three
groups (adult women, adult men, and children) are calculated. The daily exposure dose (ADD) to PAHs
through the two exposure routes is calculated as follows:

$$ADD_{der} = \frac{C \times SA \times ABS \times AF \times EF \times ED \times CF}{AT \times BW} \tag{14}$$

$$ADD_{inh} = \frac{C \times IR \times EF \times ED}{AT \times BW} \tag{15}$$

Where $ADD_{der}$ and $ADD_{inh}$ are the average daily exposure dose that enters the body through the
dermal contact and inhalation, respectively (ng kg$^{-1}$ day$^{-1}$), $C$ is the concentration of PAHs (ng m$^{-3}$). $IR$
is the inhalation rate (m$^3$ d$^{-1}$). $EF$ and $ED$ are the exposure duration (d a$^{-1}$) and period (a), respectively.
$BW$ is the body weight (kg). $SA$ is the skin exposed surface area (cm$^2$). $ABS$ is the skin absorption
factor. $AT$ is the average exposure time (d). The values are shown in Table S2.

**2.3.2. Incremental lifetime cancer risk (ILCR)**

The ILCR was calculated based on the ADD:

$$ILCR_{der} = ADD_{der} \times SFO_{der}/10^6 \tag{16}$$

$$ILCR_{inh} = ADD_{inh} \times SFO_{inh}/10^6 \tag{17}$$

$$TILCR = ILCR_{der} + ILCR_{inh} \tag{18}$$

Where $ILCR_{der}$ and $ILCR_{inh}$ are lifetime cancer risks through the dermal contact and inhalation,
respectively. $TILCR$ is the total lifetime cancer risk of exposure through the two pathways. $SFO$ is a
cancer slope factor (kg day mg$^{-1}$), and its values are shown in Table S2. For carcinogen, an $ILCR$ less
than $1 \times 10^{-6}$ indicates negligible cancer risk, an $ILCR$ between $1 \times 10^{-6}$ and $1 \times 10^{-4}$ indicates potential
cancer risk, and an $ILCR$ larger than $1 \times 10^{-4}$ indicates high potential cancer risk.



**3 Experiments setup and observation data**

**3.1 Experiments setup**

In this study, we used two nested domains covering the whole globe and East Asia as shown in Fig. S1. The horizontal resolutions are 1°× 1° and 0.33°× 0.33°, respectively. A total of 20 vertical layers are used in IAP-AACM. The first layer of the model is approximately 50 m deep and the top layer extends to 20 km. The simulation results from January 1st to December 31st 2013 and from January 1st to December 31st 2018 were used for analysis. Each simulation had a one-month spin-up before January 1st to reduce the influence of initial conditions. The global version of the Weather Research and Forecasting model (WRF, version 3.7.1) (Zhang et al., 2012a; Skamarock et al., 2008) provides the meteorological fields to drive the IAP-AACM. The initial and boundary conditions of the global WRF were produced by Final Analysis data (FNL) from the National Centers for Environmental Prediction (NCEP).

The emission inventory of BaP in 2013 and 2018 was derived from the Emissions Database for Global Atmospheric Research (EDGAR, Crippa et al., 2020, available from https://edgar.jrc.ec.europa.eu/dataset_pop60#sources, last access: 15 December 2023). We mainly analyzed the results using EDGAR emission, which mainly includes anthropogenic sources such as power, transportation, industrial, agricultural, and energy for buildings. An additional simulation for 2013 using the emission inventory developed by the research group of Peking University (PKU) (http://inventory.pku.edu.cn, last access: 10 February 2023) was used to investigate the uncertainties from emissions. The resolution of both emission inventories is 0.1°× 0.1°. Therefore, we re-gridded the emissions inventories to match the model grids at 1°× 1° and 0.33°× 0.33° resolution.



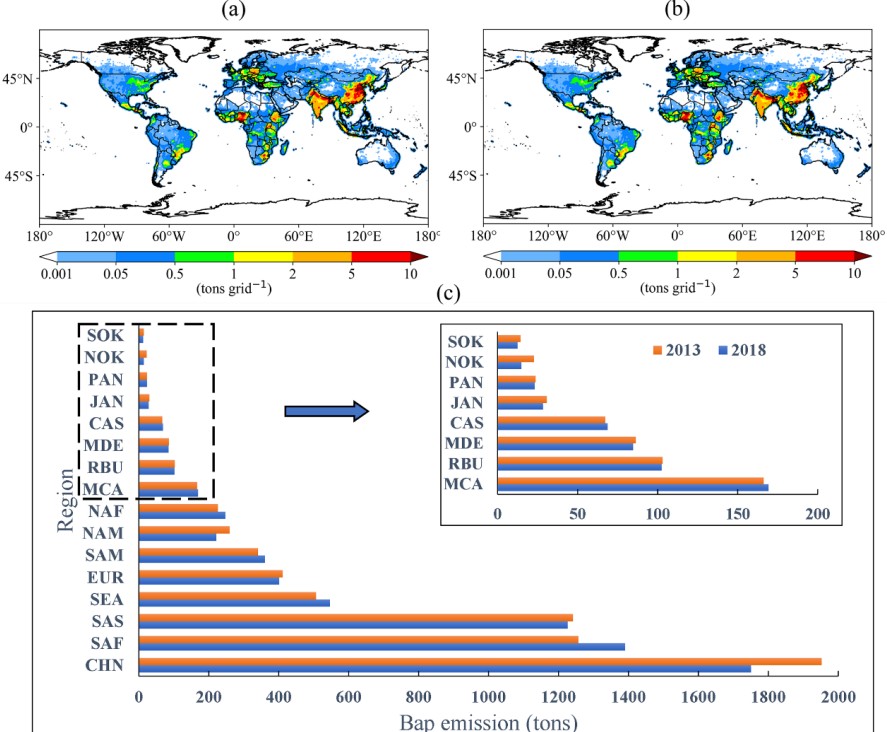


**Figure 1. Spatial distributions of total emissions of BaP in 2013 (a) and 2018 (b) based on the EDGAR**
**inventory. BaP emissions for 16 regions (except oceans, Arctic, and Antarctic) in 2013 and 2018 (c).**
The global total emissions of BaP in 2013 and 2018 are shown in Fig. 1a and 1b, respectively. The
annual emissions in different regions (Fig. S1) were also calculated (Fig. 1c). The global emissions of
BaP were 7,166.9 in 2013 and 7,109.5 t in 2018, respectively. The global emission showed a slight
decrease of (0.8%) from 2013 to 2018. China is one of the largest BaP-emitting countries in the world.
Its emissions were 1,952.2 t in 2013 and 1,750.2 t in 2018, respectively, accounting for about 27.2% and
24.6% of the world. Southern Africa and India had the second and third-largest emissions. Emissions
from China, Southern Africa, and India accounted for 62.1% and 61.4% of the world. China, Australia,
India, Europe, North America, South Korea, Japan, and North Korea displayed a declining trend from
2013 to 2018. China experienced the largest decline (10.4%) due to the active emission control measures
taken under the "Air Pollution Prevention and Control Action Plan" implemented in 2013. The emissions
increased in Africa (10.7%), South-East Asia (7.8%), and South America (5.9%).



To understand the change in BaP concentrations, we conducted five experiments: the first and
second experiments simulated the BaP concentration using the emissions in 2013 and 2018 driven by the
corresponding meteorological fields. The third experiment used the emission in 2018 but kept the
meteorological conditions in 2013 to investigate the effects of meteorological condition changes on the
concentration of BaP. Studies neglecting the heterogeneous loss of BaP and using two different
heterogeneous schemes were also performed to explore the impacts of heterogeneous reactions on BaP
concentrations in the fourth and fifth experiments
**3.2 Observational data**
To evaluate the model performance, we collected the PAHs observation from several available
datasets and more than 50 published papers. The observational data are summarized as follows: (1)
European Monitoring and Evaluation Program (EMEP, available from
https://projects.nilu.no/ccc/reports.html, last access: 15 December 2023): this includes annual and
monthly averages of BaP concentrations at 36 European sites in Spain, Finland, France, Germany,
Norway, Poland, and other countries in Europe; (2) National Air Pollution Surveillance network: (NAPS,
https://data-donnees.az.ec.gc.ca/data/air/monitor/: last access:30 January 2024): this includes daily
averages (autumn and winter) of BaP concentrations at Canadian stations; (3) Integrated Atmospheric
Deposition Network: (IADN, https://iadnviz.iu.edu/datasets/index.html, last access: 20 January 2023):
this includes monthly mean concentrations of BaP at 6 sites in the United States and Canada from 1990
to 2021; (4) Chinese Persistent Organic Pollutants (POPs) Soil and Air Monitoring Program Phase II
(SAMP-II, Ma et al., 2018): this is carried out by the International Joint Research Center for Persistent
Toxic Substances (IJRC-PTS), focusing on 11 urban centers in China (Beijing, Xi'an, Nanchang,
Kunming, Lanzhou, Chengdu, Harbin, Dalian, Lhasa, Guangzhou, and Shihezi), 1 suburb and 3
background/rural areas. This observational data only covers the period from August 2008 to July 2010;
(5) observational data collected from published papers (these sources are listed in supplementary material)
(Wu et al., 2024).
PAHs measurements data are very sparse compared to conventional pollutants (e.g., $PM_{2.5}$). Since
most of the data are not continuous in time, we selected data covering at least 10 days in years as close
as possible to the simulation year (2013) and used the mean values for comparison. The comparison of
the monthly variation was conducted only for sites in Europe where observations were continuous and





available. The locations of the BaP observation sites are shown in Fig. S2. The site information is listed
in Table S5 and Table S6.
**4 Results**
**4.1 Global distribution of BaP**

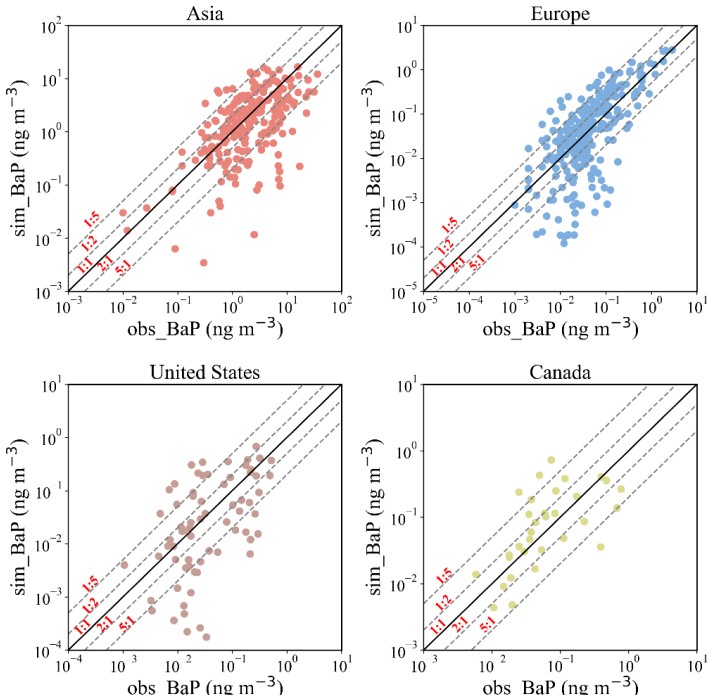

**Figure 2. Comparison of simulated (sim_BaP) and observed (obs_BaP) annual mean concentrations of BaP**
**in Asia (orange), Europe (blue), United States (brown), and Canada (green) in 2013. The solid black line**
**shows a ratio of 1 : 1 and the dashed gray lines show ratios of 5 : 1, 2 : 1, 1 : 2, and 1 : 5.**

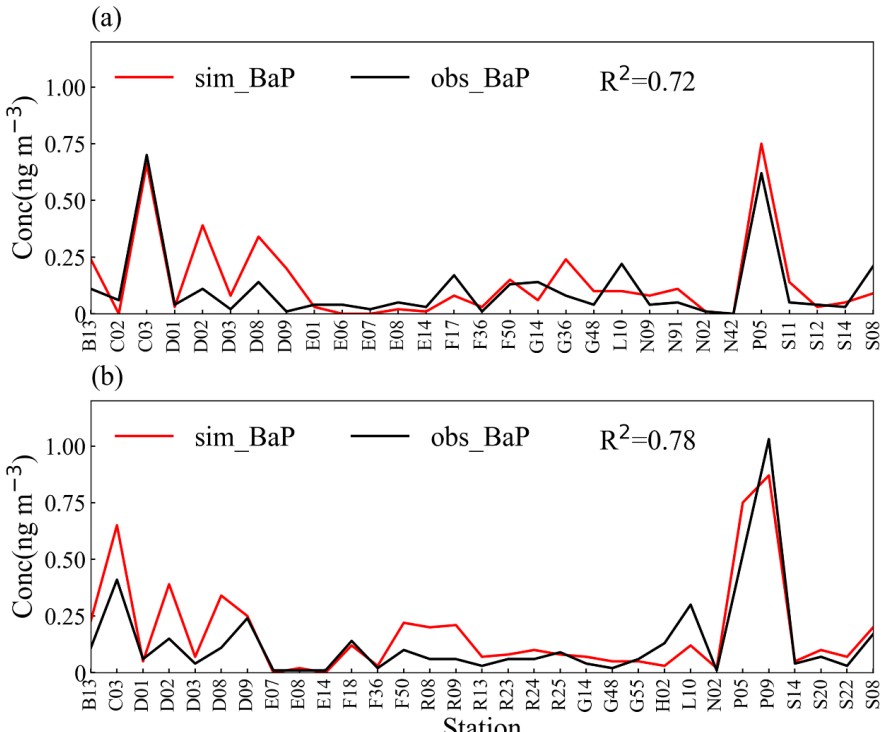

**Figure 3. Comparison of the BaP annual mean simulated (red) concentrations with observed (black) values at European sites in 2013 (a) and 2018 (b).**

To evaluate the performance of the IAP-AACM model, annual mean simulated concentrations in Asia, Europe, the United States, and Canada were compared with observations (Fig. 2). The results show that the model can reproduce nearly half of the observation samples within a factor of 2 and most observations within a factor of 5 at sites in Asia, Europe, the United States, and Canada. The number of sites where BaP was underestimated was greater than the number where it was overestimated due to the averaging effect of subgrid emissions. Considering that some of the comparisons are not in the same year, a certain discrepancy between the model and observation is expected. Further, a specific comparison was performed using the data from about 30 stations in Europe (Fig. 3a and b). High concentrations were mainly found in polluted areas of Central Europe, consistent with the simulation of Gusev et al. (2017), such as Poland (P05) and the Czech Republic (C03) with observed values of 0.70 and 0.62 ng m$^{-3}$, respectively, and simulated concentrations of 0.66 and 0.75 ng m$^{-3}$ in 2013, The model successfully reproduced the observed concentrations and differences between sites.



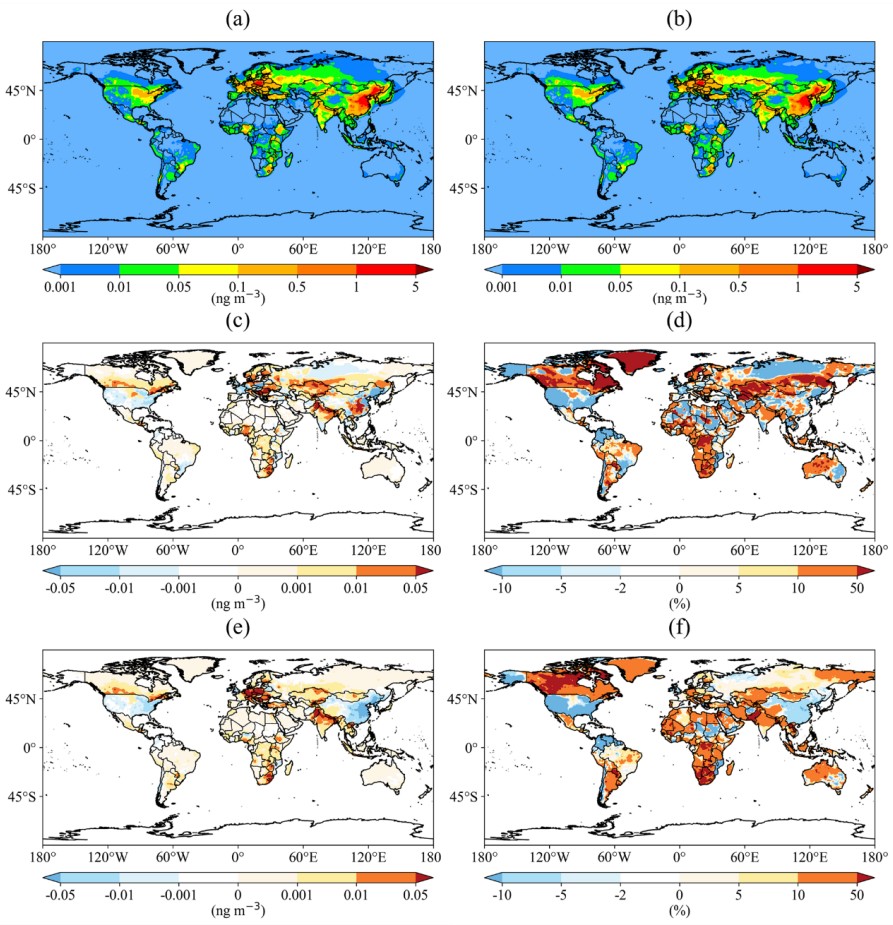

**Figure 4. Spatial distributions of annual mean BaP concentrations based on the EDGAR in 2013 (a) and 2018 (b). The absolute (c/e) and relative concentration changes (d/f) from 2013 to 2018 are shown considering both emissions and meteorological conditions (c, d) or only emissions (e,f), respectively.**

The spatial distribution of annual mean BaP concentrations based on the EDGAR inventory in 2013 and 2018 are shown in Fig. 4a and b. The spatial distribution of BaP concentrations in 2018 was similar to that in 2013. The spatial pattern was consistent with the emission distribution in the EDGAR inventory. High concentrations of BaP were found in northern and eastern China, and central Europe, even exceeding the European Union target value for BaP (1 ng m$^{-3}$), indicating an urgent need to control BaP and other PAHs. The absolute and relative concentration changes from 2013 to 2018 are shown in Fig. 4c and d. The most significant decreases were seen in Russia, the United States, eastern and northern China. By contrast, the concentration in India, Europe, Southeast Asia, and South Africa shows an





increase, with the average annual concentration increasing by 19.4%, 1.2%, 11.2%, and 18.3%,
respectively. When only considering the impact of emission change (Fig. 4e and f), the decrease in the
eastern United States is larger and the increase in central Europe is larger. In particular, there is an obvious
decline (about 8.0%) across China, which demonstrates the effect of the emission reduction measures.
These results clearly show the large influence of meteorological changes. It is crucial to consider
meteorological factors when evaluating emission changes and reduction measures through monitoring
concentrations in the atmosphere.



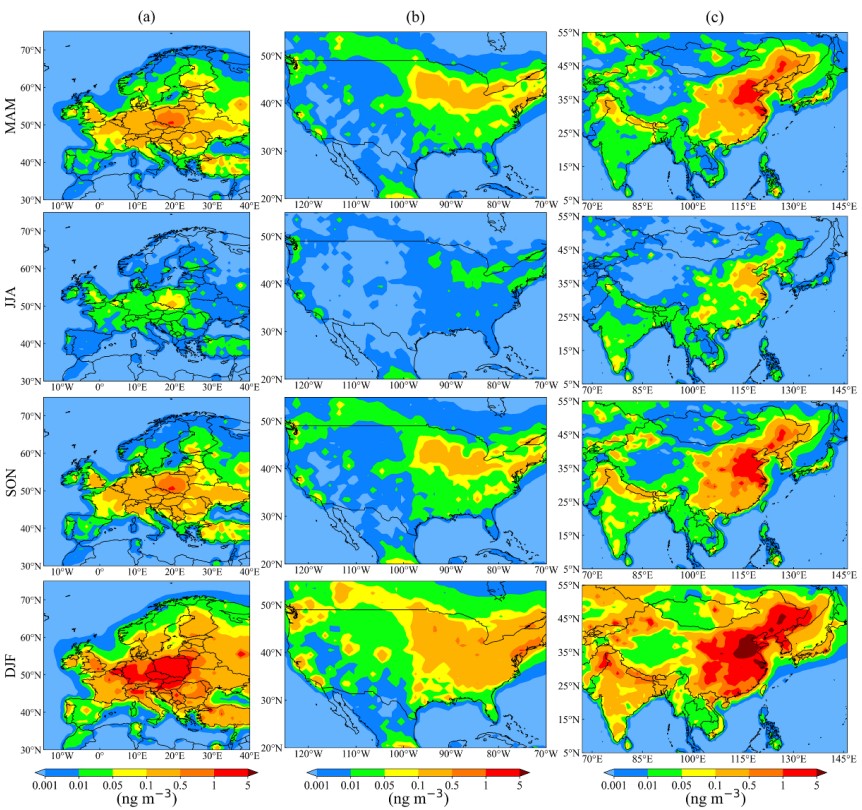

**Figure 5. Spatial distributions of seasonal mean concentrations in Europe (a), the contiguous United States (b), and East Asia (c) in 2013.**

Figure 5 shows the seasonal mean concentrations of BaP in Europe, the contiguous United States, and East Asia in four seasons: March–April–May (MAM, representing spring), June–July–August (JJA, representing summer), September–October–November (SON, representing autumn), and December–January–February (DJF, representing winter). Generally, BaP had the highest concentration in winter and lowest in summer. This is caused by the larger emission and poorer atmospheric diffusion conditions in winter than in summer. In the contiguous United States, the concentrations were lower than 1 ng m⁻³ in all four seasons, consistent with the simulation of Galarneau et al. (2014). In east China, large areas have a concentration of > 1 ng m⁻³ and even > 5 ng m⁻³ in BTH in winter. Europe shows a distribution of high values in central areas and low values in remote areas. In Central Europe (such as Poland and the Czech Republic), large areas have concentrations between 1–5 ng m⁻³ in winter. High concentrations were also reported by Bieser et al. (2012). The observation clearly shows higher concentrations at sites in Poland and the Czech Republic than at other sites in Europe (Fig. 6). The model successfully reproduces the





seasonal variation of BaP at sites in Europe. The simulation had a good agreement with observations at
C03 and P05 with correlation coefficient ($R^2$) of 0.91 and 0.91, and normalized mean bias (NMB) of -
0.04 and 0.14, respectively. The $R^2$ was higher than 0.8 at B13 and S08, and the NMB was 1.01 and -
0.55. In summary, IAP-AACM can reasonably simulate the spatial distribution and seasonal variation of
BaP.

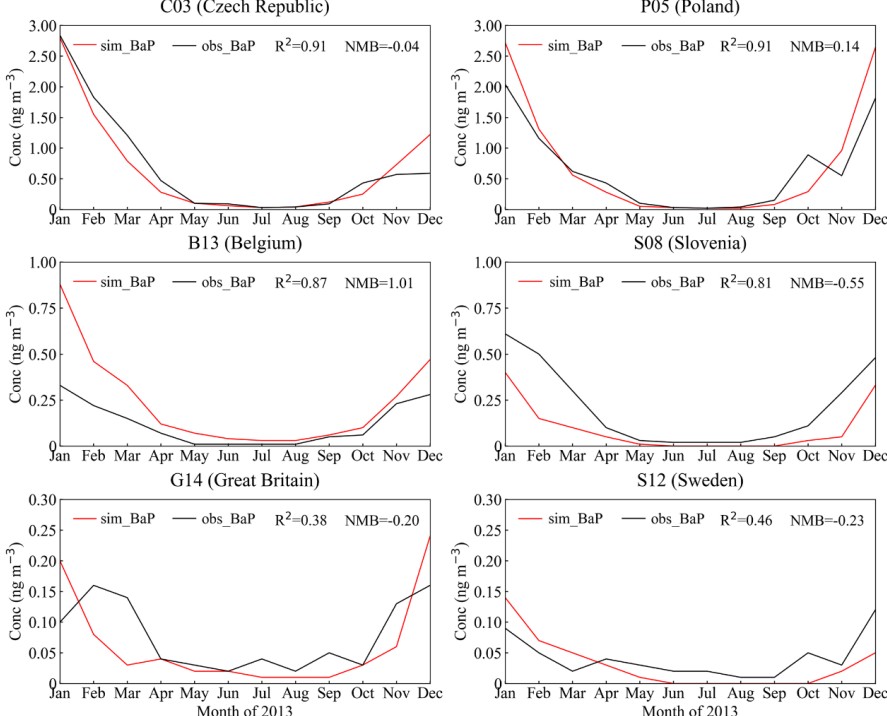


**Figure 6. Comparison of the BaP month mean simulated concentrations (red) with observed values (black)**
**at six stations in Europe in 2013.**
**4.2 Distribution of PAHs and their change in China**

Figure 7 shows the annual mean distribution of BaP in China in 2013. The concentrations ranged

from 0.02 to 6.14 ng m$^{-3}$. Overall, high concentrations were simulated in the North China Plain, East
China, and Northeast China, significantly higher than in Northwest and Southwest China, consistent with
previous studies (Ma et al., 2020; Yan et al., 2019). Among the different provinces in China, there are 14
provinces with concentrations higher than the ambient air quality standards of China (1 ng m$^{-3}$, GB 3095–
2012: http://www.zhb.gov.cn/, last access: 6 April 2014). Shanghai had the highest concentration of 6.14
ng m$^{-3}$, followed by Tianjin (4.56 ng m$^{-3}$), Beijing (3.41 ng m$^{-3}$), and Shandong (3.10 ng m$^{-3}$). The

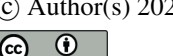



concentrations in the Northwest and Southwest regions were lower, with Tibet having the lowest
concentration of only 0.02 ng m$^{-3}$. This is due to lower levels of industrial activities and population
density in these regions compared to eastern regions. In addition, the high topography of northwest
regions has good air circulation and is conducive to the diffusion and dilution of atmospheric pollutants.
In 2013, Beijing had the highest BaP concentration in winter (14.03 ng m$^{-3}$), possibly due to the high
population density, high number of vehicles, and frequent industrial activities in Beijing. Moreover,
Beijing lies on the North China Plain, where the meteorological conditions make it easier for air
pollutants to stay and accumulate, resulting in a high concentration of BaP.

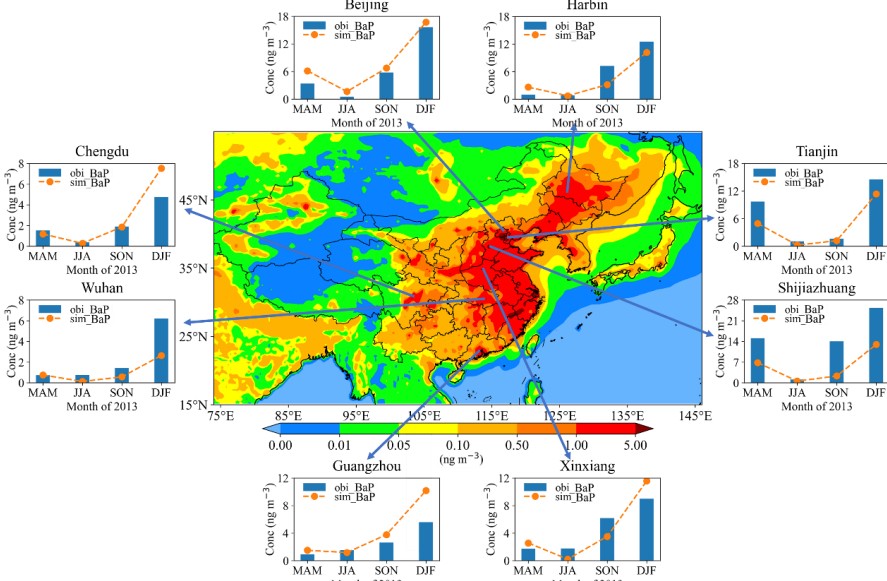

**Figure 7. Spatial distributions of annual mean concentrations in China. Comparison of the BaP month mean**
**simulated concentrations (red) with observed values (black) at eight cities in 2013.**

To reveal the seasonal variation of BaP concentrations in key regions, we analyzed the concentration

in eight major cities, i.e., Beijing, Tianjin, Shijiazhuang, Xinxiang, Wuhan, Chengdu, Guangzhou, and
Harbin (seen in Fig. 7). It can be seen that the seasonal variations of BaP in these cities are similar, with
the highest values in winter and the lowest in summer. The seasonal difference in northern cities was
significantly greater than that in southern cities. In Beijing, Xinxiang, Tianjin, Harbin, and Shijiazhuang,
the differences in concentration between winter and summer were as high as 15.06, 11.76, 11.14, 9.45,
and 12.42 ng m$^{-3}$, respectively. This is caused by the fact that coal-fired heating is very common in
northern China, which can significantly increase the PAH emissions in winter (Yan et al., 2019). In





addition, the meteorological conditions also affect the seasonal variation of PAHs, as the lower
temperature, less rainfall, and weaker solar radiation during the winter support the formation of a stable
inversion layer, greatly limiting the diffusion of BaP in the air (Lin et al., 2015; Quan et al., 2014).

By comparing the simulated concentrations with the observed concentrations, we find that the model

can capture the BaP concentrations and the seasonal pattern in different cities. For example, the observed
and simulated concentrations show good consistency in the spring, summer, and autumn of Chengdu and
in the summer, autumn, and winter of Beijing, with a deviation of only 0.04 to 1.1 ng m$^{-3}$. However, there
were some deviations between the simulated and observed concentrations. The most obvious
underestimation is seen in Shijiazhuang. This is probably due to the underestimation of emissions and
the model resolution that may not fully resolve the pollution in cities with urban areas smaller than the
model grid. The model performance could be improved by using more precise emissions and increasing
grid resolution. Nonetheless, the model can capture the magnitude and seasonal variation in BaP
concentration well in China and in other countries around the world, and can therefore be used to evaluate
the health effects of BaP exposure.

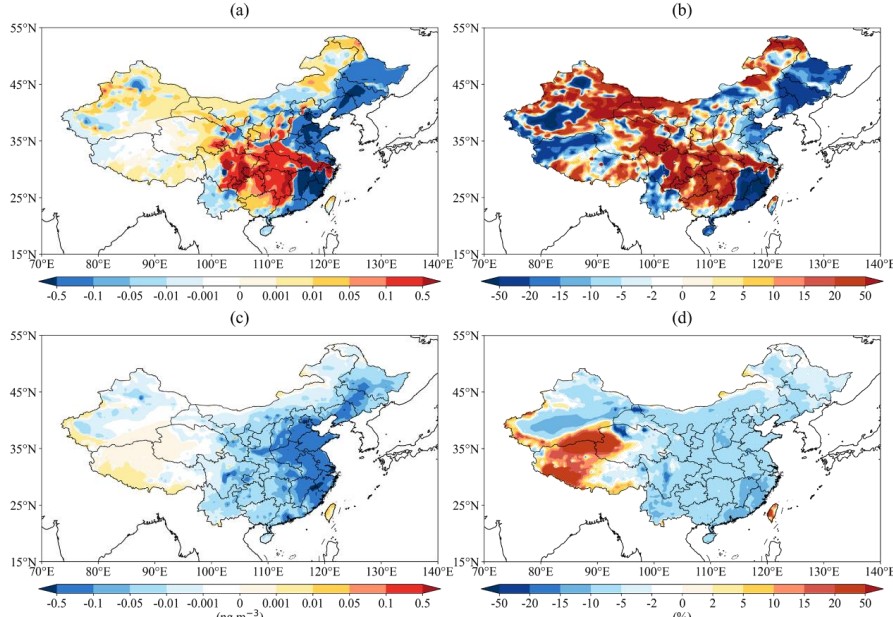

**Figure 8. The absolute (a/c) and relative concentration changes (b/d) from 2013 to 2018 in mean annual BaP**
**concentrations in China are shown considering both emissions and meteorological conditions (a, b) or only**
**emissions (c, d), respectively.**



The changes from 2013 to 2018 are shown in Fig. 8. The trend and magnitude of changes differ
greatly across different regions. The largest decrease (> 20%) was seen in northeastern and southeastern
China. The concentration also decreased in the North China Plain. The decrease was larger than the
emission reduction over these areas. By contrast, as shown in Fig. 8a, the concentration in the Sichuan
Basin showed an inverse trend although the emission decreased. This phenomenon reflects the impact of
meteorological conditions. When only considering the emissions changes, the concentration shows a
decrease over most regions consistent with the emission change. It should be noted that the decrease in
BaP in the two experiments is significantly lower than that of $PM_{2.5}$. Wang et al. (2019) showed that
compared with 2013, the concentration of $PM_{2.5}$ in the BTH, the YRD, and the PRD in 2017 decreased
by 39.6%, 34.3%, and 27.7%, respectively. For cities in North and East China, the concentration still
exceeds the national limit value (1 ng m$^{-3}$) although the concentration of BaP decreased significantly in
2018. For example, the BaP concentrations in Shanghai, Beijing, and Tianjin considering changes in both
emissions and meteorology were 5.32, 3.31, and 3.38 ng m$^{-3}$, respectively, and those with emission
changes alone were 5.58, 3.11, and 4.17 ng m$^{-3}$, indicating that the concentrations are mainly affected by
the emission sources. The results in the central and western cities differed greatly between the two
experiments, especially in Chongqing, Sichuan, and Guizhou, indicating that changes were mainly
related to meteorological conditions. Therefore, when formulating emission reduction policies, it is
necessary to take into account the effects of changes in meteorological conditions as well as emission
sources.
**4.3 Health risks of PAHs in China**



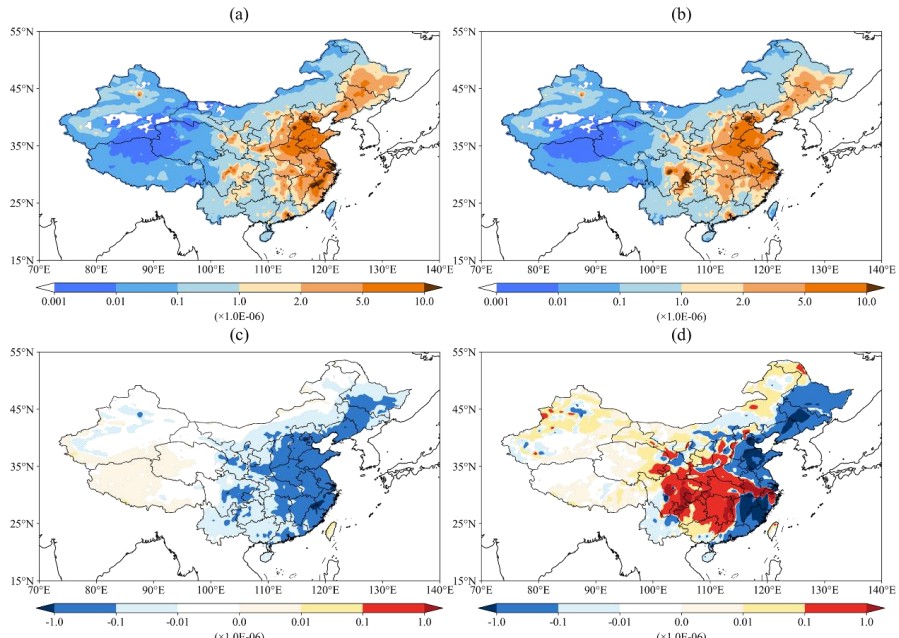

**Figure 9. The distribution of TILCR (the sum of ILCR values of the two exposure routes) in China in 2013 (a) and 2018 (b), and the absolute from 2013 to 2018 when only emissions (c) or both emissions and meteorological conditions (d) are considered.**

In this section, the health risks of BaP are assessed based on the simulation over the nested domain (domain 2) covering China. The ILCR induced by inhalation and dermal contact to BaP based on exposure for children, adult women, and adult men were calculated using Eq. (14)–Eq. (18). Figure. 9a and b show the distribution of TILCR (the sum of ILCR values of the two exposure routes) in China in 2013 and 2018, and Fig. 9c and d show the change from 2013 to 2018 when only emissions or both emissions and meteorological conditions are considered. It can be seen that the spatial distribution of TILCR (Fig. 9a) is consistent with the spatial distribution of the BaP annual concentrations (Fig. 7), showing higher risk in eastern regions than in the western regions (Han et al., 2020). The values of the TILCR in China ranged from $1.6 \times 10^{-9}$ to $3.8 \times 10^{-5}$, with an average value of $3.7 \times 10^{-7}$. Compared with 2013, the average TILCR in 2018 decreased by $3.0 \times 10^{-8}$, which is mainly directly related to the decrease in concentration. From the perspective of two exposure routes, $ILCR_{inh}$ and $ILCR_{der}$ values ranged from $1.5 \times 10^{-10}$–$3.9 \times 10^{-6}$ and $1.4 \times 10^{-9}$–$3.4 \times 10^{-5}$, with an average value of $3.9 \times 10^{-8}$ and $3.3 \times 10^{-7}$, respectively. The values of $ILCR_{der}$ were one order of magnitude higher than the $ILCR_{inh}$.



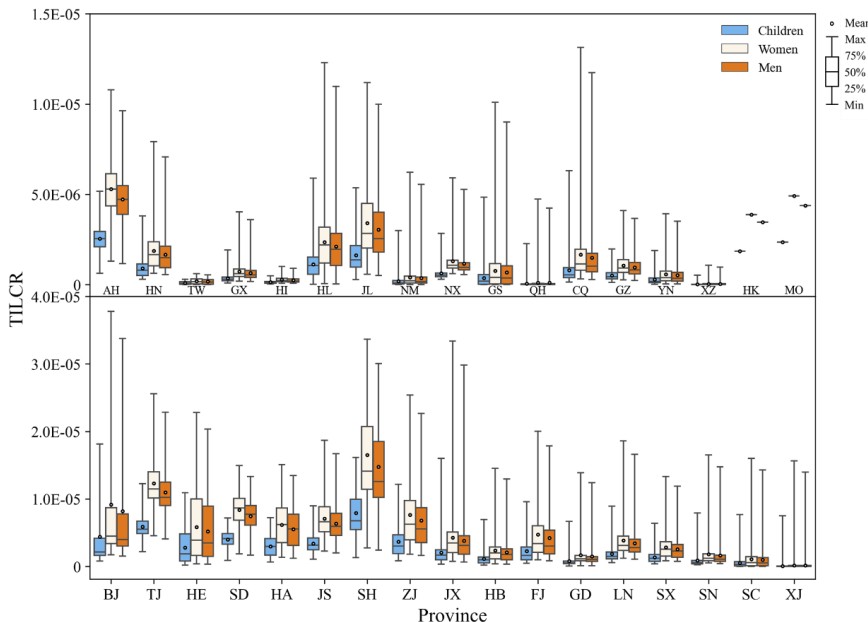

**Figure 10. The TILCR values for the three age groups (Children, Women, and Men) in different provinces of China in 2013.**

The TILCR values of the three groups in 2013 and 2018 are shown in Fig. 10 and Fig. S3 (the provinces are listed in Table S4), respectively, which ranged from $1.55 \times 10^{-9}$ to $3.78 \times 10^{-5}$ ($1.60 \times 10^{-9}$ to $3.41 \times 10^{-5}$). The order of TILCR was women ($1.46 \times 10^{-6}$) > men ($1.31 \times 10^{-6}$) > children ($7.03 \times 10^{-7}$), which was similar to that of dermal contact exposure. Overall, 29.2% of TILCR were higher than $1.0 \times 10^{-6}$, and 1.2% of TILCR were higher than $1.0 \times 10^{-5}$ in 2013. There was a slight decrease in TILCR values in 2018 due to the lower concentrations of BaP, with 27.9% and 0.7% of TILCR higher than $1.0 \times 10^{-6}$ and $1.0 \times 10^{-5}$, respectively. There is no high cancer risk in China, but there are potential cancer risks in some areas, which should be paid attention to.





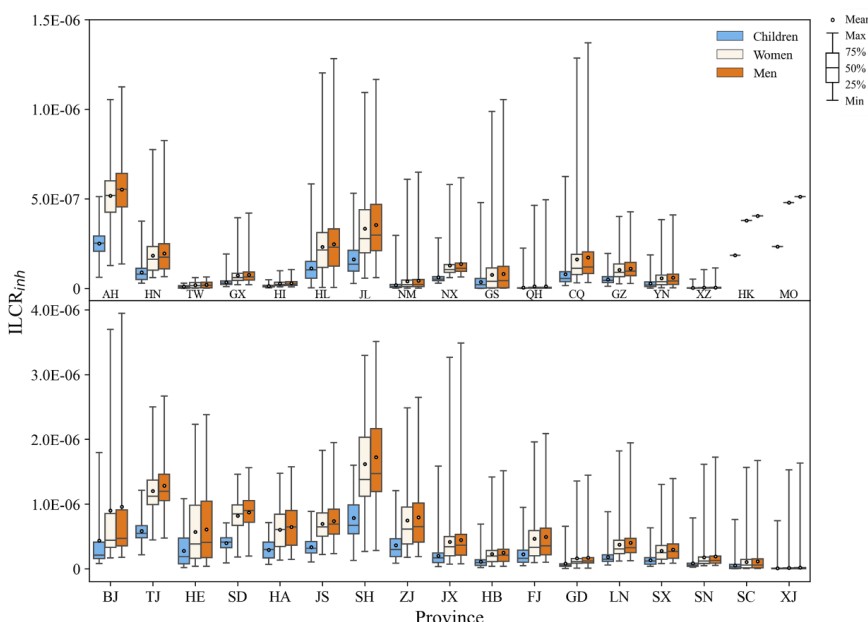

**Figure 11. The ILCR$_{inh}$ values for the three age groups (Children, Women, and Men) in different provinces of China in 2013.**

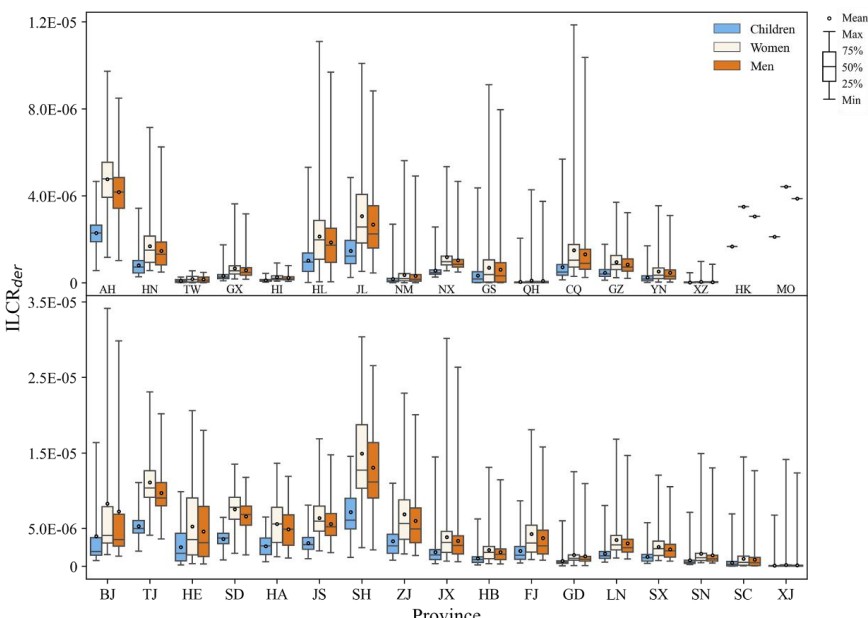

**Figure 12. The ILCR$_{der}$ values for the three age groups (Children, Women, and Men) in different provinces of China in 2013.**

The ILCR values of the three groups in China through inhalation and dermal exposure routes are



shown in Fig. 11 and Fig. 12, and the ILCR in 2018 are shown in Fig. S4 and Fig. S5. For the inhalation
pathway, the average $ILCR_{inh}$ was $1.22 \times 10^{-7}$ ($<1.0 \times 10^{-6}$). The order of $ILCR_{inh}$ was men ($1.53 \times 10^{-7}$) >
women ($1.43 \times 10^{-7}$) > children ($6.95 \times 10^{-8}$), and the risk for men was about twice that of children, but was
lower in women than in men. This may be caused by the fact that the inhalation and metabolic rate of
women are weaker than men (Bai et al., 2020). The highest average value was found in Shanghai, where
the average $ILCR_{ing}$ for the three groups were $1.72 \times 10^{-6}$, $1.62 \times 10^{-6}$, and $7.84 \times 10^{-7}$, respectively. Han et
al. (2020) found cases of excess cancer due to exposure to PAHs in large cities such as Shanghai. Only
1.6% of the three groups had $ILCR_{ing}$ higher than $1 \times 10^{-6}$, indicating that the health risks from inhalation
exposure were low. A similar conclusion was mentioned in an earlier review (Yan et al., 2019).

For dermal contact exposure, the average $ILCR_{der}$ was $1.04 \times 10^{-6}$ ($>1.0 \times 10^{-6}$). Compared to the

$ILCR_{inh}$, the health risk to adults was slightly higher than that to children, but women had higher risk
values than men. This may be caused by the fact that the body weight of women is weaker than men. The
order of $ILCR_{der}$ was women ($1.32 \times 10^{-6}$) > men ($1.15 \times 10^{-6}$) > children ($6.33 \times 10^{-6}$), which is similar to
the results of previous studies (Bai et al., 2020). Among the three groups, 27.4% of the $ILCR_{der}$ values
were higher than $1 \times 10^{-6}$, and 0.7% were higher than $1 \times 10^{-5}$. This shows that there is a greater potential
carcinogenic risk through dermal contact exposure.
**5 Discussion**

It should be noted that model results have some uncertainties even though our model simulated the

main features of PAHs concentrations reasonably well. Firstly, we simulated lower BaP concentrations
when using the PKU inventory than when using the EDGAR inventory over most continental areas,
except for Inner Mongolia, eastern Russia, and north China (Fig. S6). The difference can be as high as
0.5 ng m$^{-3}$ over some areas in wintertime although the spatial and temporal distributions are consistent.
The emission inventory remains to be constrained by more observations. Current observations are too
sparse to conduct detailed evaluation in areas where long-term measurements are not available. Secondly,
we tested the influence of heterogeneous reaction schemes on simulation. When heterogeneous reactions
were not considered, the model significantly overestimated the concentration of BaP (Fig. S7), suggesting
the importance of heterogeneous loss of BaP. Using the Langmuir-Hinshelwood mechanism, we
simulated lower concentrations in most regions of the northern hemisphere, especially in winter (Fig.



S8). However, the difference between the simulated results of the two mechanisms is significantly
reduced in summer due to high temperature and high humidity. This is consistent with the results by Mu
et al. (2018), i.e., low temperature and low humidity can significantly increase the lifetime of BaP.
Comparison of model results using different schemes and model intercomparison would further help
identify the uncertainties and improve model performance.
**6 Conclusion**

In this study, the PAHs modules were coupled into the IAP-AACM model to investigate the global

and regional distribution of PAHs. The model has the state-of -the-art heterogeneous mechanism and
allows us to consistently examine the multi-scale distribution of PAHs. Comparison with observations
shows that the model can reproduce the different concentrations of BaP at the stations in Asia, Europe,
the United States, and Canada. The model can capture the seasonal variation of BaP, with lower
concentrations in summer and higher concentrations in winter over the continents in the northern
hemisphere. The global distributions of BaP in 2013 and 2018 were very similar, with high concentrations
concentrated in eastern China and central Europe, even exceeding EU limits (1 ng m$^{-3}$). Compared with
2013, BaP concentration in 2018 showed a decrease in the United States, Poland, France, Czech, and
some regions in China. By contrast, the concentrations increased by >10% in India and South Africa.
Populations in these regions are facing increased health risks posed by PAHs.

In China, the decline in BTH (8.5%) and YRH (9.4%) benefitted from "the Action Plan". However,

the decline was significantly less than that of conventional pollutants, such as PM$_{2.5}$. Changes in
meteorological conditions had a significant influence on changes in BaP concentration, which increased
in the Sichuan Basin and central China even though the emissions over these areas decreased due to the
control measures. There was a slight decrease in total ILCR (TILCR) values in 2018 compared to 2013.
For the different exposure routes, the dermal contact was an order of magnitude higher than the inhalation
route. The TILCR for adults was greater than that for children. 29.2% of TILCR were higher than 1.0×10$^{-}$
$^{6}$, indicating that there are still potential cancer risks in China. More attention must be paid to the non-
traditional pollutant pollutants and strict but different control measures are necessary to reduce PAHs'
concentration and health risks.

In summary, our study developed an effective tool for simulating the global and regional

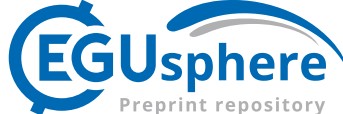



concentrations of BaP and other PAHs and quantified the health risks in China from 2013 to 2018. Model
analysis indicated that emission inventories and heterogeneous reactions can significantly affect the
simulated BaP concentrations. Accurate emissions and reasonable representation of heterogeneous
reactions would greatly reduce the gap between model results and observation. However, the current
observations are insufficient to fully evaluate and constrain the model. Especially, long-term observations
are needed in Asia, India, and Africa. These regions are still facing significant health risks. In addition,
monitoring in the background and remote regions (such as the Arctic) is necessary to quantify the long-
range transport of PAHs.

*Code and data availability.* The source code and their introduction of IAP-AACM can be found online
via Zenodo (https://doi.org/10.5281/zenodo.12214119). The simulated data can be found via Zenodo
(https://doi.org/10.5281/zenodo.11595165). All the observational data are provided in the supplement
and can be found via Zenodo (https://doi.org/10.5281/zenodo.11595165).

*Author contributions.* ZcW developed the model, prepared the input data, performed the simulations and
analysis, and wrote the paper with suggestions from all co-authors. XC supported the coding and
conceived the idea of the paper. XC and ZfW revised the paper and provided scientific guidance through
all research advances. JL, ZW, LW, HC, YL, MQ, XT, and QW modified the manuscript. WW supported
the emission data. YW, ZZ, and ZJ supported the data analysis. All listed authors have read and approved
the final manuscript.

*Competing interests.* The authors declare that they have no conflict of interest.

*Disclaimer.* Publisher's note: Copernicus Publications remains neutral about jurisdictional claims in
published maps and institutional affiliations.

*Acknowledgements.* We are particularly grateful to Prof. Oliver Wild at Lancaster University for his help
with improving the paper. We thank Prof. Alexey Gusev at EMEP for providing MSC-E results as a good
reference to test our model performance. We also thank the National Key Scientific and Technological



Infrastructure project "Earth System Science Numerical Simulator Facility" (EarthLab).

*Financial support.* This research has been supported by the National Key R&D Program of China (Grant
NO.2020YFA0607801), the National Natural Science Foundation of China (Grant NO. 42377105) and
the National Key Scientific and Technological Infrastructure project "Earth System Science Numerical
Simulator Facility".



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

Friedman, C. L.,Selin, N. E.: Long-Range Atmospheric Transport of Polycyclic Aromatic Hydrocarbons:
A Global 3-D Model Analysis Including Evaluation of Arctic Sources, Environmental Science
& Technology, 46, 9501-9510, https://doi.org/10.1021/es301904d, 2012.
Galarneau, E., Makar, P. A., Zheng, Q., Narayan, J., Zhang, J., Moran, M. D., Bari, M. A., Pathela, S.,
Chen, A.,Chlumsky, R.: PAH concentrations simulated with the AURAMS-PAH chemical
transport model over Canada and the USA, Atmospheric Chemistry and Physics, 14, 4065-4077,
https://doi.org/10.5194/acp-14-4065-2014, 2014.
Gusev, A., Ilyin, I., Rozovskaya, O., Shatalov, V., Travnikov, O., Strijkina I. Assessment of transboundary
pollution by toxic substances: Heavy metals and POPs, Meteorological Synthesizing Centre -
East, Russia, 74 pp., 2019.
Han, F. L., Guo, H., Hu, J. L., Zhang, J., Ying, Q.,Zhang, H. L.: Sources and health risks of ambient
polycyclic aromatic hydrocarbons in China, Science of the Total Environment, 698, 13,
https://doi.org/10.1016/j.scitotenv.2019.134229, 2020.
Hansen, K. M., Christensen, J. H., Brandt, J., Frohn, L. M.,Geels, C.: Modelling atmospheric transport
of α-hexachlorocyclohexane in the Northern Hemispherewith a 3-D dynamical model: DEHM-



640    POP, Atmos. Chem. Phys., 4, 1125-1137, https://doi.org/10.5194/acp-4-1125-2004, 2004.

641 Haritash, A. K.,Kaushik, C. P.: Biodegradation aspects of Polycyclic Aromatic Hydrocarbons (PAHs): A

642    review, Journal of Hazardous Materials, 169, 1-15,

643    https://doi.org/10.1016/j.jhazmat.2009.03.137, 2009.

644 Harner, T.,Bidleman, T. F.: Octanol-air partition coefficient for describing particle/gas partitioning of

645    aromatic compounds in urban air, Environmental Science & Technology, 32, 1494-1502,

646    https://doi.org/10.1021/es970890r, 1998.

647 Inomata, Y., Kajino, M., Sato, K., Ohara, T., Kurokawa, J., Ueda, H., Tang, N., Hayakawa, K., Ohizumi,

648    T.,Akimoto, H.: Source contribution analysis of surface particulate polycyclic aromatic

649    hydrocarbon concentrations in northeastern Asia by source-receptor relationships,

650    Environmental Pollution, 182, 324-334, https://doi.org/10.1016/j.envpol.2013.07.020, 2013.

651 Inomata, Y., Kajino, M., Sato, K., Ohara, T., Kurokawa, J. I., Ueda, H., Tang, N., Hayakawa, K., Ohizumi,

652    T.,Akimoto, H.: Emission and Atmospheric Transport of Particulate PAHs in Northeast Asia,

653    Environmental Science & Technology, 46, 4941-4949, https://doi.org/10.1021/es300391w,

654    2012.

655 Jonker, M. T. O.,Koelmans, A. A.: Sorption of polycyclic aromatic hydrocarbons and polychlorinated

656    biphenyls to soot and soot-like materials in the aqueous environment mechanistic considerations,

657    Environmental Science & Technology, 36, 3725-3734, https://doi.org/10.1021/es020019x, 2002.

658 Jury, W. A., Spencer, W. F.,Farmer, W. J.: Behavior Assessment Model for Trace Organics in Soil: I.

659    Model Description, Journal of Environmental Quality, 12, 558-564,

660    https://doi.org/10.2134/jeq1983.00472425001200040025x, 1983.

661 Kahan, T. F., Kwamena, N. O. A.,Donaldson, D. J.: Heterogeneous ozonation kinetics of polycyclic

662    aromatic hydrocarbons on organic films, Atmospheric Environment, 40, 3448-3459,

663    https://doi.org/10.1016/j.atmosenv.2006.02.004, 2006.

664 Karickhoff, S. W.: Semi-empirical estimation of sorption of hydrophobic pollutants on natural sediments

665    and soils, Chemosphere, 10, 833-846, https://doi.org/10.1016/0045-6535(81)90083-7, 1981.

666 Keyte, I. J., Harrison, R. M.,Lammel, G.: Chemical reactivity and long-range transport potential of

667    polycyclic aromatic hydrocarbons - a review, Chemical Society Reviews, 42, 9333-9391,

668    https://doi.org/10.1039/c3cs60147a, 2013.



Klöpffer, W., Wagner, B.,Scheringer, M.: Atmospheric degradation of organic substances data for
persistence and long-range transport potential, Environmental Science and Pollution Research
- International, 14, 143-144, https://doi.org/10.1065/espr2007.04.408, 2007.
Kwamena, N. O. A., Clarke, J. P., Kahan, T. F., Diamond, M. L.,Donaldson, D. J.: Assessing the
importance of heterogeneous reactions of polycyclic aromatic hydrocarbons in the urban
atmosphere using the Multimedia Urban Model, Atmospheric Environment, 41, 37-50,
https://doi.org/10.1016/j.atmosenv.2006.08.016, 2007.
Lammel, G., Dvorská, A., Klánová, J., Kohoutek, J., Kukacka, P., Prokes, R.,Sehili, A. M.: Long-range
Atmospheric Transport of Polycyclic Aromatic Hydrocarbons is Worldwide Problem - Results
from Measurements at Remote Sites and Modelling, Acta Chimica Slovenica, 62, 729-735, 2015.
Lammel, G.,Sehili, A. M.: Global fate and distribution of polycyclic aromatic hydrocarbons emitted from
Europe and Russia, Atmospheric Environment, 41, 8301-8315,
https://doi.org/10.1016/j.atmosenv.2007.06.050, 2007.
Lammel, G., Sehili, A. M., Bond, T. C., Feichter, J.,Grassl, H.: Gas/particle partitioning and global
distribution of polycyclic aromatic hydrocarbons--a modelling approach, Chemosphere, 76, 98-
106, https://doi.org/10.1016/j.chemosphere.2009.02.017, 2009.
Li, J., Wang, Z. F., Zhuang, G., Luo, G., Sun, Y.,Wang, Q.: Mixing of Asian mineral dust with
anthropogenic pollutants over East Asia: a model case study of a super-duststorm in March 2010,
Atmospheric Chemistry and Physics, 12, 7591-7607, https://doi.org/10.5194/acp-12-7591-2012,
2012.

Li, R. F., Zhang, J.,Krebs, P.: Global trade drives transboundary transfer of the health impacts of
polycyclic aromatic hydrocarbon emissions, Communications Earth & Environment, 3, 13,
https://doi.org/10.1038/s43247-022-00500-y, 2022.
Li, Z., Mulholland, J. A., Romanoff, L. C., Pittman, E. N., Trinidad, D. A., Lewin, M. D.,Sjödin, A.:
Assessment of non-occupational exposure to polycyclic aromatic hydrocarbons through
personal air sampling and urinary biomonitoring, Journal of Environmental Monitoring, 12,
1110-1118, https://doi.org/10.1039/c000689k, 2010.
Lin, Y., Ma, Y. Q., Qiu, X. H., Li, R., Fang, Y. H., Wang, J. X., Zhu, Y. F.,Hu, D.: Sources, transformation,
and health implications of PAHs and their nitrated, hydroxylated, and oxygenated derivatives in



PM2.5 in Beijing, Journal of Geophysical Research-Atmospheres, 120, 7219-7228,

https://doi.org/10.1002/2015jd023628, 2015.

Liu, S. J., Lu, Y. L., Wang, T. Y., Xie, S. W., Jones, K. C.,Sweetman, A. J.: Using gridded multimedia

model to simulate spatial fate of Benzo α pyrene on regional scale, Environment International,

63, 53-63, https://doi.org/10.1016/j.envint.2013.10.015, 2014.

Lou, S. J., Shrivastava, M., Ding, A. J., Easter, R. C., Fast, J. D., Rasch, P. J., Shen, H. Z., Simonich, S.

704          M., Smith, S. J., Tao, S.,Zelenyuk, A.: Shift in Peaks of PAH-Associated Health Risks From

East Asia to South Asia and Africa in the Future, Earths Future, 11, 15,

https://doi.org/10.1029/2022ef003185, 2023.

Ma, W. L., Liu, L. Y., Jia, H. L., Yang, M.,Li, Y. F.: PAHs in Chinese atmosphere Part I: Concentration,

source and temperature dependence, Atmospheric Environment, 173, 330-337,

https://doi.org/10.1016/j.atmosenv.2017.11.029, 2018.

Ma, W. L., Zhu, F. J., Liu, L. Y., Jia, H. L., Yang, M.,Li, Y. F.: PAHs in Chinese atmosphere Part II: Health

risk assessment, Ecotoxicology and Environmental Safety, 200, 9,

https://doi.org/10.1016/j.ecoenv.2020.110774, 2020.

Mu, Q., Shiraiwa, M., Octaviani, M., Ma, N., Ding, A. J., Su, H., Lammel, G., Pöschl, U.,Cheng, Y. F.:

Temperature effect on phase state and reactivity controls atmospheric multiphase chemistry and

transport of PAHs, Science Advances, 4, 8, https://doi.org/10.1126/sciadv.aap7314, 2018.

Nam, K. J., Li, Q., Heo, S. K., Tariq, S., Loy-Benitez, J., Woo, T. Y.,Yoo, C. K.: Inter-regional multimedia

fate analysis of PAHs and potential risk assessment by integrating deep learning and climate

change scenarios, Journal of Hazardous Materials, 411, 12,

https://doi.org/10.1016/j.jhazmat.2021.125149, 2021.

Octaviani, M., Tost, H.,Lammel, G.: Global simulation of semivolatile organic compounds - development

and evaluation of the MESSy submodel SVOC (v1.0), Geoscientific Model Development, 12,

3585-3607, https://doi.org/10.5194/gmd-12-3585-2019, 2019.

Odabasi, M., Cetin, E.,Sofuoglu, A.: Determination of octanol–air partition coefficients and supercooled

liquid vapor pressures of PAHs as a function of temperature: Application to gas–particle

partitioning in an urban atmosphere, Atmospheric Environment, 40, 6615-6625,

https://doi.org/10.1016/j.atmosenv.2006.05.051, 2006.



Quan, J. N., Tie, X. X., Zhang, Q., Liu, Q., Li, X., Gao, Y.,Zhao, D. L.: Characteristics of heavy aerosol
pollution during the 2012-2013 winter in Beijing, China, Atmospheric Environment, 88, 83-89,
https://doi.org/10.1016/j.atmosenv.2014.01.058, 2014.
Ravindra, K., Sokhi, R.,Van Grieken, R.: Atmospheric polycyclic aromatic hydrocarbons: Source
attribution, emission factors and regulation, Atmospheric Environment, 42, 2895-2921,
https://doi.org/10.1016/j.atmosenv.2007.12.010, 2008.
San José, R., Pérez, J. L., Callén, M. S., López, J. M.,Mastral, A.: BaP (PAH) air quality modelling
exercise over Zaragoza (Spain) using an adapted version of WRF-CMAQ model,
Environmental Pollution, 183, 151-158, https://doi.org/10.1016/j.envpol.2013.02.025, 2013.
Seigneur, C., Karamchandani, P., Lohman, K., Vijayaraghavan, K.,Shia, R. L.: Multiscale modeling of
the atmospheric fate and transport of mercury, Journal of Geophysical Research-Atmospheres,
106, 27795-27809, https://doi.org/10.1029/2000jd000273, 2001.
Semeena, V. S.,Lammel, G.: The significance of the grasshopper effect on the atmospheric distribution
of   persistent   organic   substances,   Geophysical   Research   Letters,   32,   5,
https://doi.org/10.1029/2004gl022229, 2005.
Shen, H. Z., Tao, S., Liu, J. F., Huang, Y., Chen, H., Li, W., Zhang, Y. Y., Chen, Y. C., Su, S., Lin, N., Xu,
Y. Y., Li, B. G., Wang, X. L.,Liu, W. X.: Global lung cancer risk from PAH exposure highly
depends on emission sources and individual susceptibility, Scientific Reports, 4, 8,
https://doi.org/10.1038/srep06561, 2014.
Shrivastava, M., Lou, S., Zelenyuk, A., Easter, R. C., Corley, R. A., Thrall, B. D., Rasch, P. J., Fast, J. D.,
Simonich, S. L. M., Shen, H. Z.,Tao, S.: Global long-range transport and lung cancer risk from
polycyclic aromatic hydrocarbons shielded by coatings of organic aerosol, Proceedings of the
National Academy of Sciences of the United States of America, 114, 1246-1251,
https://doi.org/10.1073/pnas.1618475114, 2017.
Skamarock, W. C., Klemp, J. B., Dudhia, J., Gill, D., Barker, D. M., Duda, M. G., Huang, X.-Y., Wang,
752            W.,Powers, J. G. A Description of the Advanced Research WRF Version 3. 2008.
Smith, R. L., Davis, J. M.,Speckman, P.: Assessing the human health risk of atmospheric particles,
Novartis   Found   Symp,   220,   59-72;   discussion   72-9,
https://doi.org/10.1002/9780470515600.ch4, 1999.



Stockwell, W. R., Middleton, P., Chang, J. S.,Tang, X. Y.: The Second Generation Regional Acid
Deposition Model Chemical Mechanism for Regional Air Quality Modeling, Journal of
Geophysical          Research-Atmospheres,          95,          16343-16367,
https://doi.org/10.1029/JD095iD10p16343, 1990.

Strand, A.,Hov, O.: A model strategy for the simulation of chlorinated hydrocarbon distributions in the
global    environment,    Water    Air    and    Soil    Pollution,    86,    283-316,
https://doi.org/10.1007/bf00279163, 1996.

Su, C., Zheng, D. F., Zhang, H.,Liang, R. Y.: The past 40 years' assessment of urban-rural differences in
Benzo a pyrene contamination and human health risk in coastal China, Science of the Total
Environment, 901, 9, https://doi.org/10.1016/j.scitotenv.2023.165993, 2023.

van Noort, P. C. M.: A thermodynamics-based estimation model for adsorption of organic compounds by
carbonaceous materials in environmental sorbents, Environmental Toxicology and Chemistry,
22, 1179-1188, 2003.

Walcek, C. J.,Aleksic, N. M.: A simple but accurate mass conservative, peak-preserving, mixing ratio
bounded advection algorithm with Fortran code, Atmospheric Environment, 32, 3863-3880,
https://doi.org/10.1016/s1352-2310(98)00099-5, 1998.

Wang, L., Zhang, F. Y., Pilot, E., Yu, J., Nie, C. J., Holdaway, J., Yang, L. S., Li, Y. H., Wang, W. Y.,
Vardoulakis, S.,Krafft, T.: Taking Action on Air Pollution Control in the Beijing-Tianjin-Hebei
(BTH) Region: Progress, Challenges and Opportunities, International Journal of Environmental
Research and Public Health, 15, 27, https://doi.org/10.3390/ijerph15020306, 2018.

Wang, Y. S., Li, W. J., Gao, W. K., Liu, Z. R., Tian, S. L., Shen, R. R., Ji, D. S., Wang, S., Wang, L. L.,
Tang, G. Q., Song, T., Cheng, M. T., Wang, G. H., Gong, Z. Y., Hao, J. M.,Zhang, Y. H.: Trends
in particulate matter and its chemical compositions in China from 2013-2017, Science China-
Earth Sciences, 62, 1857-1871, https://doi.org/10.1007/s11430-018-9373-1, 2019.

Wang, Z. F., Maeda, T., Hayashi, M., Hsiao, L. F.,Liu, K. Y.: A nested air quality prediction modeling
system for urban and regional scales: Application for high-ozone episode in Taiwan, Water Air
and Soil Pollution, 130, 391-396, https://doi.org/10.1023/a:1013833217916, 2001.

Wang, Z. X., Li, J. X., Mu, X., Zhao, L. Y., Gu, C., Gao, H., Ma, J. M., Mao, X. X.,Huang, T.: A WRF-
CMAQ modeling of atmospheric PAH cycling and health risks in the heavy petrochemical



industrialized Lanzhou valley, Northwest China, Journal of Cleaner Production, 291, 9,
https://doi.org/10.1016/j.jclepro.2021.125989, 2021.

Wei, Y., Chen, X. S., Chen, H. S., Li, J., Wang, Z. F., Yang, W. Y., Ge, B. Z., Du, H. Y., Hao, J. Q., Wang,
788        W., Li, J. J., Sun, Y. L.,Huang, H. L.: IAP-AACM v1.0: a global to regional evaluation of the
atmospheric chemistry model in CAS-ESM, Atmospheric Chemistry and Physics, 19, 8269-
8296, https://doi.org/10.5194/acp-19-8269-2019, 2019.

Wu, Z. C., Chen, X. S., and Wang, Z. F.: A Global-Regional Nested Model of Polycyclic aromatic
hydrocarbons, Zenodo [code], https://doi.org/10.5281/zenodo.12214119, 2024.

Wu, Z. C., Chen, X. S., and Wang, Z. F.: Results and validation of Global-Regional Nested Model for
polycyclic aromatic hydrocarbons., Zenodo [data set],
https://doi.org/10.5281/zenodo.11595165, 2024.

Yan, D. H., Wu, S. H., Zhou, S. L., Tong, G. J., Li, F. F., Wang, Y. M.,Li, B. J.: Characteristics, sources
and health risk assessment of airborne particulate PAHs in Chinese cities: A review,
Environmental Pollution, 248, 804-814, https://doi.org/10.1016/j.envpol.2019.02.068, 2019.

Ye, Q., Li, J., Chen, X., Chen, H., Yang, W., Du, H., Pan, X., Tang, X., Wang, W., Zhu, L., Li, J., Wang,
Z.,Wang, Z.: High-resolution modeling of the distribution of surface air pollutants and their
intercontinental transport by a global tropospheric atmospheric chemistry source–receptor
model (GNAQPMS-SM), Geoscientific Model Development, 14, 7573-7604,
https://doi.org/10.5194/gmd-14-7573-2021, 2021.

Zaveri, R. A.,Peters, L. K.: A new lumped structure photochemical mechanism for large-scale
applications, Journal of Geophysical Research-Atmospheres, 104, 30387-30415,
https://doi.org/10.1029/1999jd900876, 1999.

Zhang, L., Brook, J. R.,Vet, R.: A revised parameterization for gaseous dry deposition in air-quality
models, Atmospheric Chemistry and Physics, 3, 2067-2082, https://doi.org/10.5194/acp-3-
2067-2003, 2003.

Zhang, M., Xie, J. F., Wang, Z. T., Zhao, L. J., Zhang, H.,Li, M.: Determination and source identification
of priority polycyclic aromatic hydrocarbons in PM2.5 in Taiyuan, China, Atmospheric
Research, 178, 401-414, https://doi.org/10.1016/j.atmosres.2016.04.005, 2016.

Zhang, Q., Zheng, Y. X., Tong, D., Shao, M., Wang, S. X., Zhang, Y. H., Xu, X. D., Wang, J. N., He, H.,





Liu, W. Q., Ding, Y. H., Lei, Y., Li, J. H., Wang, Z. F., Zhang, X. Y., Wang, Y. S., Cheng, J., Liu,

Y., Shi, Q. R., Yan, L., Geng, G. N., Hong, C. P., Li, M., Liu, F., Zheng, B., Cao, J. J., Ding, A.

816             J., Gao, J., Fu, Q. Y., Huo, J. T., Liu, B. X., Liu, Z. R., Yang, F. M., He, K. B.,Hao, J. M.: Drivers

of improved PM2.5 air quality in China from 2013 to 2017, Proceedings of the National

Academy of Sciences of the United States of America, 116, 24463-24469,

https://doi.org/10.1073/pnas.1907956116, 2019.

Zhang, Y., Hemperly, J., Meskhidze, N.,Skamarock, W. C.: The Global Weather Research and

Forecasting (GWRF) Model: Model Evaluation, Sensitivity Study, and Future Year Simulation,

Atmospheric and Climate Sciences, 2, 231-253, https://doi.org/10.4236/acs.2012.23024, 2012a.

Zhang, Y., Jaeglé, L., van Donkelaar, A., Martin, R. V., Holmes, C. D., Amos, H. M., Wang, Q., Talbot,

R., Artz, R., Brooks, S., Luke, W., Holsen, T. M., Felton, D., Miller, E. K., Perry, K. D., Schmeltz,

D., Steffen, A., Tordon, R., Weiss-Penzias, P.,Zsolway, R.: Nested-grid simulation of mercury

over North America, Atmospheric Chemistry and Physics, 12, 6095-6111,

https://doi.org/10.5194/acp-12-6095-2012, 2012b.

Zhang, Y., Shen, H., Tao, S.,Ma, J.: Modeling the atmospheric transport and outflow of polycyclic

aromatic hydrocarbons emitted from China, Atmospheric Environment, 45, 2820-2827,

https://doi.org/10.1016/j.atmosenv.2011.03.006, 2011a.

Zhang, Y., Tao, S., Ma, J.,Simonich, S.: Transpacific transport of benzo[a]pyrene emitted from Asia,

Atmospheric Chemistry and Physics, 11, 11993-12006, https://doi.org/10.5194/acp-11-11993-

2011, 2011b.

Zhang, Y. X.,Tao, S.: Global atmospheric emission inventory of polycyclic aromatic hydrocarbons (PAHs)

for 2004, Atmospheric Environment, 43, 812-819,

https://doi.org/10.1016/j.atmosenv.2008.10.050, 2009.

Zhang, Y. X., Tao, S., Shen, H. Z.,Ma, J. M.: Inhalation exposure to ambient polycyclic aromatic

hydrocarbons and lung cancer risk of Chinese population, Proceedings of the National Academy

of Sciences of the United States of America, 106, 21063-21067,

https://doi.org/10.1073/pnas.0905756106, 2009.

Zhen, Z. X.: Observation and simulation of atmospheric polycyclic aromatic hydrocarbons in the North

China Plain, Ph.D.thesis, Nanjing university of information science and technology, China, 142



843   pp., 2023.

844  Zhu, F. J., Ma, W. L., Hu, P. T., Zhang, Z. F.,Li, Y. F.: Temporal trends of atmospheric PAHs: Implications

845   for the influence of the clean air action, Journal of Cleaner Production, 296, 8,

846   https://doi.org/10.1016/j.jclepro.2021.126494, 2021.