# Peer review of "Modeling of PAHs From Global to Regional Scales: Model Development (IAP-AACM\_PAH v1.0) and Investigation of Health Risks in 2013 and 2018 in China"

_EGUsphere, 2024_

## Author Comment (AC1)

Dear Editors and Reviewers,

Thank you for your valuable comments on our manuscript entitled "Modeling of PAHs From Global to Regional Scales: Model Development and Investigation of Health Risks from 2013 to 2018 in China" (MS No.: EGUSPHERE-2024-1437). Those comments are greatly helpful for improving our manuscript. We provided point-by-point replies to your comments and revised the manuscript accordingly (in **blue and highlighted**).

The comments (in black) are copied here and responses to the comments (in blue) are as follows:

**Response to Referee #1:**

This study employed an improved global-regional nested Atmospheric Aerosol and Chemistry Model (IAP-AACM) to investigate the global distribution of PAHs and the health risk in 2013 and 2018, respectively. The new developed model includes more parameterized processes and shows satisfied simulation results. The results reveal a decline of PAH concentration in most area of China except Sichuan Basin, attributed to meteorology conditions. However, the total incremental lifetime cancer risk posed by BaP only show a slight decrease and the health risks still exist especially in East China. All the findings indicate more tough control measures for PAH when considering both pollution and public health. Besides, meteorology factors play an important role when assessing the control measures for concentration measuring.

The paper is well structured and contributes importance for air pollution and health risks. However, there are several limitations outlined below that need to be addressed before considering it for publication.

Response: We appreciate your comments and suggestions, and we have revised the manuscript accordingly.

**Comment 1:** The title as well as main text of this paper shows the analysis results "from 2013 to 2018", while the simulation tests were performed for 2013 and 2018,

respectively. Therefore, the expression is ambiguous.

**Response:** We agree with this point. We have changed the expression "from 2013 to 2018" to "in 2013 and 2018". (Line3)

**Comment 2:** In abstract, Line 28-29, the study concluded the decrease for BaP is smaller than PM2.5 during the same period. However, it seems no analysis was performed in section4 to support this conclusion.

**Response:** Thanks for the comment. The relevant description has been added in Section 4. (Line 457-458)

**Comment 3:** Is the computation for TILCR consider the population data for each simulation resolution? From formulation 14-18, there are no population data related but in Line 214-216, the population data were used.

**Response:** We apologize for this confusion. We considered the population data with different resolutions when calculating the TILCR values, as described in the manuscript (Line 219) "re-gridded to 1° x 1° and 0.33° x 0.33° to match the model resolution". According to this comment, we also added the description of population database resolution in the revised manuscript. (Line 217)

**Comment 4:** 9 shows the distribution of TILCR. As shown in formula (18), the TILCR is calculated for children, women and men, respectively. Is the TILSR in Fig.9 calculated with the number of children, women and men as weight? If so, please clarify how to obtain the TILCR in this figure.

**Response:** Yes, the meaning of TILCR is not clear in Fig. 9 in the original manuscript. The TILCR in Fig. 9 is the total lifetime risk of cancer through dermal contact and inhalation exposure, where we have averaged the inhalation rate (IR), exposure duration (ED), body weight (BW), and surface area of skin exposure (SA) for children, women and men to calculate the TILCR. It has to be admitted that this method would slightly overestimate the TILCR for children and underestimate the TILCR for men and women due to insufficient consideration of age and gender differences. However, these

uncertainties are acceptable according to the studied using this method (Nam et al., 2021; Su et al., 2023). To clarify the meaning of TILCR, we added the following sentence in the revised manuscript "TILCR (the sum of ILCR values of the two exposure routes after averaging the parameters of the different groups)". (Line 385-386)

**Comment 5:** Line 436-437, "It can be seen that the spatial distribution of TILCR (Fig. 9a) is consistent with the spatial distribution of the BaP annual concentrations". According to the formula (14)-(18), the TILCR seems to be proportional with the concentration of PAHs, as the other parameters have fixed values, so the TILCR should be consistent with Fig.7. The same problem also exists regard to Fig.10. The differences between children, women and men depends on the coefficients in formula (14)-(17). In my understanding, the conclusion can be obtained directly from these formulas. Please elaborate more about the meaning of these two figures.

**Response:** We completely agree with this point. However, the health risks are classified as negligible, potential, and high potential depending on the concentrations, and we believe that the health risks posed by the different routes to different regions and groups can be demonstrated concretely through Fig. 9-12. Combined with the comment 6, we added two figures showing the health risks grade distribution for a more intuitive understanding (Fig. R1).

**Comment 6:** In part 4.3, the author wants to show the health risks of PAHs. I think it's better to add a figure of the distribution of health risk grade in China for a more intuitive understanding.

**Response:** Thank you for your constructive comments. As shown in Fig. R1a and b, the health risks in western China is negligible, while there is a potential cancer risk in eastern China. The figures have been added in the revised manuscript (Figure 10).

[Figure]

Figure R1. The distribution of health risks grade in China in (a) 2013 and (b) 2018, the distribution of TILCR in (c) 2013 and (d) 2018, and the TILCR changes from 2013 to 2018 when considering the change in (c) emissions only, (f) both emissions and meteorological conditions. This figure corresponds to Figure 10 in the revised manuscript.

**Comment 7:** Formulation (14) lacks the explanation for parameter AF and CF.

**Response:** Thank you for the careful review. The explanation and values for AF and CF have been provided in the revised manuscript and supplement (Table S2). (Line 228, 236)

**Comment 8:** Line 507, it should be YRD, not YRH.

**Response:** It has been corrected. (Line 550).

**Comment 9:** 7, the figure annotation has some errors. Simulated concentrations are in orange not red and observed values are in blue not lack, please check.

**Response:** "Red" and "black" have been changed to "orange" and "blue" in Line 418.

**References**

Nam, K. J., Li, Q., Heo, S. K., Tariq, S., Loy-Benitez, J., Woo, T. Y., Yoo, C. K.: Inter-regional multimedia fate analysis of PAHs and potential risk assessment by integrating deep learning and climate change scenarios, Journal of Hazardous Materials, 411, 12, https://doi.org/10.1016/j.jhazmat.2021.125149, 2021.

Su, C., Zheng, D. F., Zhang, H., Liang, R. Y.: The past 40 years' assessment of urban-rural differences in Benzo a pyrene contamination and human health risk in coastal China, Science of the Total Environment, 901, 9, https://doi.org/10.1016/j.scitotenv.2023.165993, 2023.

---

## Author Comment (AC2)

Dear Editors and Reviewers,

Thank you for your valuable comments on our manuscript entitled "Modeling of PAHs From Global to Regional Scales: Model Development and Investigation of Health Risks from 2013 to 2018 in China" (MS No.: EGUSPHERE-2024-1437). Those comments are greatly helpful for improving our manuscript. We provided point-by-point replies to your comments and revised the manuscript accordingly (in **blue and highlighted**).

The comments (in black) are copied here and responses to the comments (in blue) are as follows:

**Response to Referee #2:**

The study investigates the global distribution of polycyclic aromatic hydrocarbons (PAHs) and their health risks, focusing on China from 2013 to 2018. Using the IAP-AACM model, the authors evaluate the spatial and seasonal distribution of BaP, a key PAH indicator. The authors show BaP concentrations have decreased in China due to emission reductions but increased in India and Southern Africa. Despite the reductions, the health risks in China, particularly in East China, remain significant.

By developing a PAH module, the study offers significant insights into PAH distribution and health risks, providing valuable insights into PAH trends across developing regions. I am happy to see its publication in due course. I would like to suggest some minor revisions after addressing the following questions:

**Response:** Thank you so much for your constructive comments and insightful suggestions. We have revised the paper carefully.

**Comment 1:** The authors should justify the capability of the EDGAR inventory to capture the short-term emission changes between 2013-2018 at the regional scale.

**Response:** Thank you for your valuable comments. The emission inventories of BaP in this paper were derived from the Emissions Database for Global Atmospheric Research (EDGAR) and Peking University (PKU). According to statistical data, China was one

of the largest PAH-emitting countries in both inventories, accounting for about 27.2% and 29.6% of the world emission in 2013, respectively. Africa and South Asia had the second and third-largest emissions, with residential combustion accounting for 87.3% and 84.0% of total emissions in 2013, respectively. This is related to the widespread use of biomass fuels for heating and cooking in developing countries.

We further calculated the changes in BaP emissions between 2013 and 2018 based on EDGAR inventory and found that BaP emission in China decreased by 10.4%, which is consistent with Wang et al. (2021) (PAH emissions decreased by 11.36% from 2013 to 2017). The results of the change in emissions from different sectors showed that the industrial sector contributed the most to the decline of BaP emissions, followed by the residential combustion, which decreased by 18.9% and 5.1% from 2013 to 2018, respectively, (Figure R1), Wang et al. (2021) also showed that emissions from the industrial and residential/commercial sectors declined by 17.32% and 10.58% from 2013 to 2017.

The simulated concentration and their change are also an indication of the emissions as the concentration is largely determined by emission. Compared to 2013, the simulated concentrations of BaP in 2018 showed a decreasing trend in Germany, Poland, Czech Republic (in Europe), Chicago, Sturgeon Point, and Cleveland (in United States), which is the same as the results from observational datasets (EMEP and IADN). In contrast, the concentrations of BaP in developing countries such as Africa and India showed an increasing trend, which is related to high population growth rates lead to increased residential combustion and energy consumption activities in developing countries (Han et al., 2022). In addition, the PKU emission data showed a similar change in BaP emission. Overall, the EDGAR inventory has a reasonable emissions distribution and the capacity to capture short-term emissions changes relatively well. According to your suggestions, we have added a figure for emissions and changes in some regions and relevant descriptions in the supplement and revised manuscript to show the ability of EDGAR inventory to capture the short-term emission changes. (Line 265-267, 269-274, and 279-286)

[Figure]

Figure R1. Emissions and changes for different sectors in China, Africa, South Asia, Europe, North America, and Sorth America in 2013 and 2018. This figure corresponds to Figure S3 in the revised manuscript.

**Comment 2:** Figure 8 looks like just a zooming in of Figure 4 with a focus over China. Again, how to verify the very local changes in Figure 8?

**Response:** We apologize for the lack of clarity giving rise to misunderstanding. Figure 8 is not just a zooming-in of Fig. 4. Actually, we used a multi-scale domain-nesting approach to simulate BaP over China. Figure 4 shows the global spatial distributions of annual mean BaP concentrations at low resolution (1° × 1°), and Fig. 8 shows the regional distribution at fine resolution (0.33° × 0.33°). Figure R2 showed the absolute (Fig. R2a and 2c) and relative concentration changes (Fig. R2b and 2d) at different resolutions in China. According to your suggestion, we have revised the colorbar of Fig. 4c-f to be consistent with Fig. 8, making the difference more visible.

Furthermore, we agree with you that it is important to verify the regional changes. The results of the model are affected by uncertainties from the emission inventory, and yet the observations in China are very sparse and most of the data are not continuous in

time, which makes it very difficult to validate the results of the changes in China. Compared with the references (Line 447-448 and 452-454), we found that the simulated concentration of BaP reproduced the declining trend in Beijing (Lin et al., 2024), Shanghai (Yang et al., 2021), Shenyang (Zhang et al., 2023), Tianjin (Zhang et al., 2022), and Chengdu (Xue et al., 2024). The concentration of BaP showed an increase in the Sichuan basin when meteorological conditions were considered. This may be related to the decrease in temperature and planetary boundary layer height compared to 2013, which is unfavorable for pollutant diffusion (Ding et al., 2019). In summary, our simulation results can basically reflect the regional variations of the BaP concentrations.

[Figure]

Figure R2. The (a/c) absolute and (b/d) relative concentration changes (b/d) of annual mean BaP concentrations (a, b) at low resolution and (c, d) fine-resolution in China are shown, respectively.

**Comment 3:** In Figures10-12, it is not easy for the readers to understand the abbreviation of each province.

**Response:** We completely agree with this point. It is a bit difficult and inconvenient for the reader to understand abbreviations for each province in Fig. 10-12. As shown in the Figure below, we have used the full name of the province instead of the abbreviation in

Fig. 10-12 and deleted the text "the provinces are listed in Table S4)". In addition, Fig. S4-S6 has also been modified in the supplement.

[Figure]

Figure R3. The TILCR values for the three age groups (Children, Women, and Men) in different provinces of China in 2013. This figure corresponds to Figure 11 in the revised manuscript.

**Comment 4:** I am wondering what's the motivation to focus on the PAH health risks over China, and it is possible to include more developing regions?

**Response:** In the paper, we mainly focused on the PAHs health risks in China for two reasons: Firstly, as air quality in China has been improved dramatically, it is necessary to assess the impact of "the Action Plan on Air Pollution Prevention and Control" promulgated by the State Council of China in 2013 on the concentration of BaP in the atmosphere and the health risks. Secondly, according to Eq. (14)-Eq. (18), the health risk is positively correlated with the concentration of BaP in the atmosphere. As can be

seen from the spatial distribution of annual mean BaP concentrations (Fig. 4 in the manuscript), the BaP concentrations in China are significantly higher than those in other developing regions.

Thanks for your good suggestion from a global perspective, we have also added a figure (Fig. 7) and the description of the global distribution of health risks in Sect.4.1 in the revised manuscript (Line385-398). As can be seen in Figure R4, the most of the countries (except China) just face negligible cancer risk, and no regions facing high potential cancer risk when evaluation is based on annual mean concentration. However, it should be noted that TILCR values increased in 2018 compared to 2013. For example, the highest TILCR in Africa in 2018 is twice as high as 2013 and changed from negligible to potential cancer risk. These countries will be likely to face potential even high potential cancer risks, especially in winter. In addition, Lou et al. (2023) has indicated that the peaks of health risks will likely shift from East Asia in 2008 to South Asia and Africa by 2050, due to the increase of residential fuels use along with the population growth in these regions. Therefore, it is clear that clean development is a necessary consideration for developing countries to avoid the health risks posed by PAHs.

[Figure]

Figure R4. The distribution of health risks grade in (a) 2013 and (b) 2018, the distribution of TILCR in (c) 2013 and (d) 2018, and the absolute from 2013 to 2018 when considering the change in (e) emissions only, (f) both emissions and meteorological conditions. This figure corresponds to Figure 7 in the revised manuscript.

**Comment 5:** In terms of PAH control, I suggest to add more policy-related discussions. For example, more specific measures could be proposed.

**Response:** This is a very good point. The policy-related discussions are added and specific measures are proposed for PAH control based on our analysis (Line 89-92, 411-412, 469-470, and 596-601).

**Line 89-92:** "This Action Plan established many effective emission reduction and energy-saving policies, such as strengthening industrial emission standards, eliminating outdated polluting industries, upgrading industrial boilers, and developing clean fuels in the residential sector".

**Line 411-412:** "This can be largely attributed to the industrial and residential coal combustion in these regions".

**Line 469-470:** "The concentration also decreased in the North China Plain, benefitting from the emission reduction policies such as the development of cleaner energy sources and the control of industrial emissions".

**Line 596-601:** "Especially in the fall and winter seasons, the concentration of BaP and the associated health risk is significantly higher than in other seasons. Management efforts on key sectors (e.g., industrial and residential sources) should be further strengthened. In heavily polluted cities, using clean energy to replace coal combustion, adjusting the energy structure of factory production, and developing innovative technologies with lower and even no emissions would be helpful to reduce PAHs pollution".

**References**

Ding, D., Xing, J., Wang, S.X., Liu, K.Y., Hao, J.M.: Estimated Contributions of Emissions Controls, Meteorological Factors, Population Growth, and Changes in Baseline Mortality to Reductions in Ambient PM2.5 and PM2.5-Related Mortality in China, 2013-2017. Environmental Health Perspectives, 127, 12, https://doi.org/10.1289/ehp4157, 2019.

Han, F.L., Kota, S. H., Sharma, S., Zhang, J., Ying, Q., Zhang, H. L.: Modeling polycyclic aromatic hydrocarbons in India: Seasonal variations, sources and associated health risks. Environmental Research, 212, 14, https://doi.org/10.1016/j.envres.2022.113466, 2022.

Lin, Y., Shi, X. D., Qiu, X. H., Jiang, X., Liu, J. M., Zhong, P. W., Ge, Y. H., Tseng, C.-H., Zhang, J. F., Zhu, T., Araujo, J. A., Zhu, Y. F.: Reduction in polycyclic aromatic hydrocarbon exposure in Beijing following China's clean air actions.

Science Bulletin, https://doi.org/10.1016/j.scib.2024.08.015, 2024.

Lou, S.J., Shrivastava, M., Ding, A.J., Easter, R.C., Fast, J.D., Rasch, P.J., Shen, H.Z., Simonich, S.M., Smith, S.J., Tao, S., Zelenyuk, A.: Shift in Peaks of PAH-Associated Health Risks From East Asia to South Asia and Africa in the Future. Earths Future, 11, 15, https://doi.org/10.1029/2022ef003185, 2023.

Wang, T., Li, B.J., Liao, H., Li, Y.: Spatiotemporal distribution of atmospheric polycyclic aromatic hydrocarbon emissions during 2013-2017 in mainland China. Science of The Total Environment, 789, 10, https://doi.org/10.1016/j.scitotenv.2021.148003, 2021.

Xue, Q.Q., Tian, Y.Z., Song, D.L., Huang, F.X., Feng, Y.C.: Variations of source-specific risks for inhalable particles-bound PAHs during long-term air pollution controls in a Chinese megacity: Impact of gas/particle partitioning. Atmos. Environ., 331, 11, https://doi.org/10.1016/j.atmosenv.2024.120565, 2024.

Yang, L., Zhang, X., Xing, W.L., Zhou, Q.Y., Zhang, L.L., Wu, Q., Zhou, Z.J., Chen, R.J., Toriba, A., Hayakawa, K., Tang, N.: Yearly variation in characteristics and health risk of polycyclic aromatic hydrocarbons and nitro-PAHs in urban shanghai from 2010-2018. Journal of Environmental Sciences, 99, 72-79, https://doi.org/10.1016/j.jes.2020.06.017, 2021.

Zhang, H., Zhang, X., Wang, Y., Bai, P.C., Zhang, L.L., Chen, L.J., Han, C., Yang, W.J., Wang, Q.M., Cai, Y.P., Nagao, S., Tang, N.: Factor analysis of recent variations of atmospheric polycyclic aromatic hydrocarbons (PAHs) and 1-nitropyrene (1-NP) in Shenyang, China from 2014 to 2020. Atmospheric Pollution Research, 14, 8, https://doi.org/10.1016/j.apr.2023.101900, 2023.

Zhang, J.W., Feng, L.H., Zhao, Y., Hou, C.C., Gu, Q.: Health risks of PM2.5-bound polycyclic aromatic hydrocarbon (PAH) and heavy metals (PPAH&HM) during the replacement of central heating with urban natural gas in Tianjin, China. Environ. Geochem. Health, 44, 2495-2514, https://doi.org/10.1007/s10653-021-01040-8, 2022.

---

## Author Comment (AC3)

**Response to Editor:**

**Comment 1:** The main paper must give the model name and version number (or other unique identifier) in the title.

**Response:** Thanks for your reminder. Considering the PAH module coupled into the IAP-AACM v1.0 model (Wei et al., 2019) in this paper, the title is modified to "Modeling of PAHs From Global to Regional Scales: Model Development (IAP-AACM_PAH v1.0) and Investigation of Health Risks in 2013 and 2018 in China" in the revised manuscript (seen in the title in the revised manuscript).

**References**

Wei, Y., Chen, X.S., Chen, H.S., Li, J., Wang, Z.F., Yang, W.Y., Ge, B.Z., Du, H.Y., Hao, J.Q., Wang, W., Li, J.J., Sun, Y.L., Huang, H.L.: IAP-AACM v1.0: a global to regional evaluation of the atmospheric chemistry model in CAS-ESM. Atmospheric Chemistry and Physics, 19, 8269-8296, https://doi.org/10.5194/acp-19-8269-2019, 2019.

---

## Author Response (AR2)

**Response to Editor:**

 **Comment 1:** Some of your figures may contain a territory that is disputed according to the United Nations. If and when the manuscript is accepted for final revised publication, you will be asked to choose one of the following options: (a) you could remove the disputed territory from the maps and submit new figure files, or (b) we could add a statement that some figures contain disputed territories.

**Response:** Thanks for your reminder. Authors chose to (b) add a statement that some figures contain disputed territories, but we are unclear in which section of the manuscript it would be more appropriate to place the statement. We are looking forward to your guidance.

On behalf of all authors, best regards,

Xueshun Chen